# Beyond cysts – organization of epithelial networks in the murine thymus

Stepan Vodopyanov[1,2], Leslie Gunther-Cummins[3,4], Sophia DesMarais[1], Maria K. Lagou[1], Xheni Nishku[3], Joseph Churaman[3], Hillary Guzik[3,5], Rotem Alon[3,5], Vera DesMarais[3,4,5], Frank Macaluso[3,4,5] and George S. Karagiannis[1,2,5,6,7,8,*]

## ABSTRACT

The thymus originates from the third pharyngeal pouch endoderm, which also gives rise to respiratory tract elements. Here, we examined intrathymic cystic structures, long considered remnants of organogenesis. Through sequential histology and ultrastructural imaging, we uncovered that these 'cysts' are in fact continuous and structured epithelial networks embedded within the thymic parenchyma. These networks follow a conserved 'head-neck-funnel-tentacle' architecture spanning the trabeculae, cortex, corticomedullary junction (CMJ) and medulla. The head, typically glandular and ciliated, connects to a funnel enriched in diverse epithelial cell types – goblet, tuft, club, ionocyte-like, microfold and ciliated cells – at the CMJ. Tentacle-like projections sometimes extend into the medulla, often surrounding perivascular spaces. Luminal contents vary, with thymocytes and macrophages most abundant caudally. We also identified solitary medullary thymic epithelial cells with large ciliated cytoplasmic lumens, distinct from these epithelial networks. Electron microscopy suggested a respiratory identity and thymic-specific adaptations for the lining cells. These findings challenge the notion of thymic cysts as inert debris, and instead reveal a coherent, mimetic system with possible roles in thymocyte selection, maturation and egress.

KEY WORDS: Thymus, Respiratory epithelium, Ciliated cells, Serial immunofluorescence, Transmission electron microscopy, 3D array tomography

## INTRODUCTION

The thymus, derived from the endoderm of the third pharyngeal pouch, is a central lymphoid organ essential for the development of a diverse and self-tolerant T cell repertoire (Hilfer and Brown, 1984; Ohigashi et al., 2016; Su et al., 2001; Wallin et al., 1996; Zou et al., 2006). Within its lobular architecture, the endoderm differentiates into highly specialized epithelial compartments, each orchestrating distinct stages of thymocyte maturation. Cortical thymic epithelial cells (cTECs), located in the subcapsular and cortical zones, are responsible for attracting early thymocyte progenitors (ETPs) and initiating lineage commitment through chemokine signaling (e.g. CCL25 and CXCL12) and Notch ligands. cTECs facilitate positive selection, allowing thymocytes with appropriately low-affinity T-cell receptors (TCRs) for self-peptide-major histocompatibility complex (MHC) to survive. Medullary thymic epithelial cells (mTECs), by contrast, mediate central tolerance through autoimmune regulator (AIRE)-dependent expression of tissue-restricted antigens, promoting deletion of autoreactive thymocytes and induction of regulatory T cells (Tregs). This compartmentalized epithelial system enables the thymus to serve its dual function, that is, producing competent T cells while preventing autoimmunity (Bhalla et al., 2022; Marx et al., 2021; Takahama et al., 2017; Wang et al., 2019).

Beyond these well-characterized epithelial subsets, the thymus also contains enigmatic cystic structures, variably described as ducts, acini, alveoli or cysts (Dooley et al., 2005a; Kirkman and Kirkman-Liff, 1985; Nabarra and Andrianarison, 1986; Shier, 1981). Historically, these elements have been viewed as developmental debris, that is remnants of the thymopharyngeal duct (Dooley et al., 2005a). These cystic profiles often express characteristics of respiratory epithelium, such as ciliated columnar cells and goblet cells, reinforcing the notion of an embryological linkage to the respiratory system (Dooley et al., 2005a). However, growing evidence has challenged this interpretation. Emerging studies on mimetic populations, epithelial cells within the thymus that phenocopy peripheral tissues, suggest that these respiratory-like cells might serve functional roles, such as presenting non-thymic self-antigens to developing thymocytes (Cabric and Brown, 2023; Givony et al., 2023; Michelson et al., 2022; Michelson and Mathis, 2022). Some models propose that these epithelial cysts might participate in negative selection or support thymocyte egress, yet these hypotheses remain speculative due to limited anatomical and functional characterization (Michelson and Mathis, 2022).

During preadolescence, the thymus generates a crucial lifelong T cell repertoire that supports cancer immune surveillance, central tolerance and systemic immunity (Wang et al., 2020). These functions typically decline with increasing age, obesity and other physiological processes, and as such, the pediatric thymus is key for establishment of long-lasting immune competence (Lagou and Karagiannis, 2023; Minato et al., 2020). In addition, the thymus displays heightened sensitivity to cytotoxic agents, resulting in acute thymic involution (ATI) (Alawam et al., 2020; Ansari and Liu, 2017; Kinsella and Dudakov, 2020). For example, cytoablative chemotherapy, which is

[1]Department of Microbiology and Immunology, Albert Einstein College of Medicine, Bronx, NY 10461, USA. [2]Integrated Imaging Program for Cancer Research, Montefiore-Einstein Comprehensive Cancer Center, Bronx, NY 10461, USA. [3]Analytical Imaging Facility, Albert Einstein College of Medicine, Bronx, NY 10461, USA. [4]Department of Cell Biology, Albert Einstein College of Medicine, Bronx, NY 10461, USA. [5]Gruss-Lipper Biophotonics Center, Albert Einstein College of Medicine, Bronx, NY 10461, USA. [6]Cancer Dormancy Institute, Montefiore-Einstein Comprehensive Cancer, Center, Bronx, NY 10461, USA. [7]Marilyn and Stanley M. Katz Institute for Immunotherapy for Cancer and Inflammatory Disorders, Montefiore-Einstein Comprehensive Cancer Center, Bronx, NY 10461, USA. [8]Tumor Microenvironment and Metastasis Program, Montefiore-Einstein Comprehensive Cancer Center, Bronx, NY 10461, USA.

*Author for correspondence (georgios.karagiannis@einsteinmed.edu)

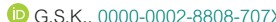 G.S.K., 0000-0002-8808-707X

often used for treating pediatric malignancies, induces both structural and functional detriments to thymic homeostasis and systemic immunity (Basta-Kaim et al., 2001; Fletcher et al., 2009; Lagou et al., 2022). Although the thymus possesses an inherent capacity for self-regeneration, certain chemotherapies severely compromise the cTEC and mTEC network, thus obfuscating the pace and quality of endogenous regeneration in both pediatric cancer mouse models and human patients (Alawam et al., 2020; Ansari and Liu, 2017; Kinsella and Dudakov, 2020). Although emerging data have documented the impact of cytoablative stressors on thymic infrastructure, including thymic epithelial cells, vessels and mesenchyme, the regenerative processes and the specific effects of such stressors on these enigmatic epithelial structures also remain largely unexplored.

Given these unresolved questions, this study undertook a systematic re-evaluation of the morphology, cellular composition and spatial organization of organized intrathymic epithelial networks. Although historically considered developmental remnants, it remains unclear whether these structures serve inert roles or participate in functional epithelial networks that support thymic homeostasis. To address this, we moved beyond isolated histological observations, aiming to reconstruct their broader topological context and explore their potential contributions to thymic architecture and epithelial diversity. Importantly, we also investigated how chemotherapeutic injury, an underexplored but clinically relevant stressor, affects these structures. Because chemotherapy profoundly disrupts thymic stromal integrity and immune regeneration, understanding its impact on these enigmatic epithelial domains may reveal novel insights into thymic resilience and post-injury repair mechanisms.

## RESULTS AND DISCUSSION
### Organizational structure of intrathymic epithelial networks
Based on morphology, different types of thymic cavities containing elements reminiscent of respiratory epithelium have been previously described, resembling glands, ducts, tubules, cysts, alveoli or acini (Arnesen and Kierulf, 1961; Clark, 1963; Isler, 1976; Kirkman and Kirkman-Liff, 1985; Shier, 1981). We have characterized more than 400 such cavities distributed across 33 murine thymi, approximately half of which had been subjected to chemotherapy to induce thymic involution (see Materials and Methods), and confirmed the presence of all the above (or similar) histological profiles (Fig. 1; Figs S1–S6). Previous studies conducted over 30–50 years ago often examined these cavities in isolation, using either vehicle-treated or drug-involuted thymi. However, via interval cutting of sequential sections, here, we demonstrate that thymic cavities observed in different fields-of-view frequently interconnect across the sequential planes-of-view, forming intricate labyrinthine networks within the thymic parenchyma (Fig. 1; Figs S1–S6).

The intrathymic epithelial networks comprise diverse and, at times transitional, components that extend from the trabeculae and outer cortex to the corticomedullary junction (CMJ) and deep medulla (Fig. 1; Figs S1–S6). Their lining cells resemble those of the respiratory epithelium in the trachea, bronchi, bronchioles and distal airways (Fig. 1G), with context-specific patterns varying by thymic location. Owing to their extreme morphological variability, it is nearly impossible to capture all features within a single network through serial sections. Therefore, this study includes seven distinct networks, collectively representing both common and unique structural characteristics (Tables 1, 2).

To assess the degree of structural organization, we analyzed the epithelial networks from a contextual perspective, starting with those located rostrally in trabecular regions and moving caudally. The rostral part of the network, referred to as the 'head', is

predominantly glandular and resides within the terminal tips of the trabeculae, which can extend to the level of the CMJ (Fig. 1H-J). These glandular elements frequently connect to the CMJ through a narrow 'neck' portion, which typically exhibits tubular or ductal morphology. Upon reaching the CMJ, the epithelial network extends laterally for significant distances, sometimes dividing into multiple branches, resulting in a 'funnel-like' structure resembling one or multiple cystic cavities. In most cases, these networks terminate abruptly with blunt ends within the CMJ. Occasionally, smaller channels, termed 'tentacles', lined primarily by squamous epithelial cells, branch out from a funnel portion and extend deeper into the medulla. These tentacles, representing the most caudal parts of the epithelial networks, tend to form smaller cystic cavities resembling alveolar structures. An example of a complete epithelial network following these underlying principles is shown in Fig. 1. For this network, five standalone planes-of-view from the same field-of-view (Fig. 1A–E) containing a total of 13 distinguished cavities (Fig. 1H–T), comprising a head portion (Fig. 1H–J), a neck portion (Fig. 1K–M), a funnel portion (Fig. 1N–R) and a tentacle portion (Fig. 1S,T), were co-registered. This analysis demonstrates the interconnection of these epithelial cavities following image reconstruction (Fig. 1F), thus providing a key for their organizational principles, as described above (Fig. 1G). Overall, these general features are either wholly or partially observed in other intrathymic epithelial networks and variants (Figs S1–S6).

### The head and neck portions of intrathymic epithelial networks
The rostral part of the epithelial network, referred to as the 'head', is located at the terminal tips of trabeculae (Fig. 1A–F,H–J; Figs S1A–E, S2A–J, S3A,J, S4A–K) and exhibits significant variation in size. In many cases, the head forms an elongated glandular or tubular structure that extends a considerable distance within and along the trabeculum (Fig. 1A–E,K–M; Figs S2A–Y, S4A–F). In yet other instances, it is much shorter, invading into the cortex to form the neck portion of the epithelial network (Fig. S2A–E,H–O). In chemotherapy-involuted thymi, the head portion remains a dominant component of the epithelial network; however, it is more simplified, often appearing as a shorter profile, consistent with the widespread cortical impairment (Fig. S3B,I).

The principal lining cells of the head portion are tall columnar cells with basal nuclei, characteristic of a secretory phenotype (Fig. 1I; Figs S1F–T, S2F–J, S3H, S4G–K). Many of these cells exhibit cilia pointing to the lumen of the head (Figs S1K,P,R, S3H, O, S4G,J), whereas a small subset of cells interspersed within the epithelial wall contain distinct mucous droplets, reflecting a goblet cell population (Fig. S3H,O). Certain sections are stained using immunofluorescence for epithelial keratins KRT5 and KRT8, which are commonly used in thymic pathology to distinguish between cTECs (KRT5$^-$KRT8$^+$) and mTECs (KRT5$^+$KRT8$^+$) (Colic et al., 1989; Farr and Braddy, 1989; Gupta et al., 2016; Lee et al., 2011; Loning et al., 1982). These markers also serve to differentiate between terminally differentiated cells and progenitor populations in respiratory epithelia (Cheng et al., 2024; Halldorsson et al., 2007; Nagle et al., 1985). As expected, the majority of cells in the head portion are KRT8$^{hi}$KRT5$^{lo}$ (Fig 1H,J; Figs S1G,I, S4K), consistent with their terminal differentiation into the columnar phenotype. The respiratory epithelia are also known to embed basal cells, that is triangular or polygonal cells that sit on the basal lamina of the respiratory cavity but lack direct contact with the lumen (Davis and Wypych, 2021). This basal cell population is believed to function as the progenitor for other respiratory cell types

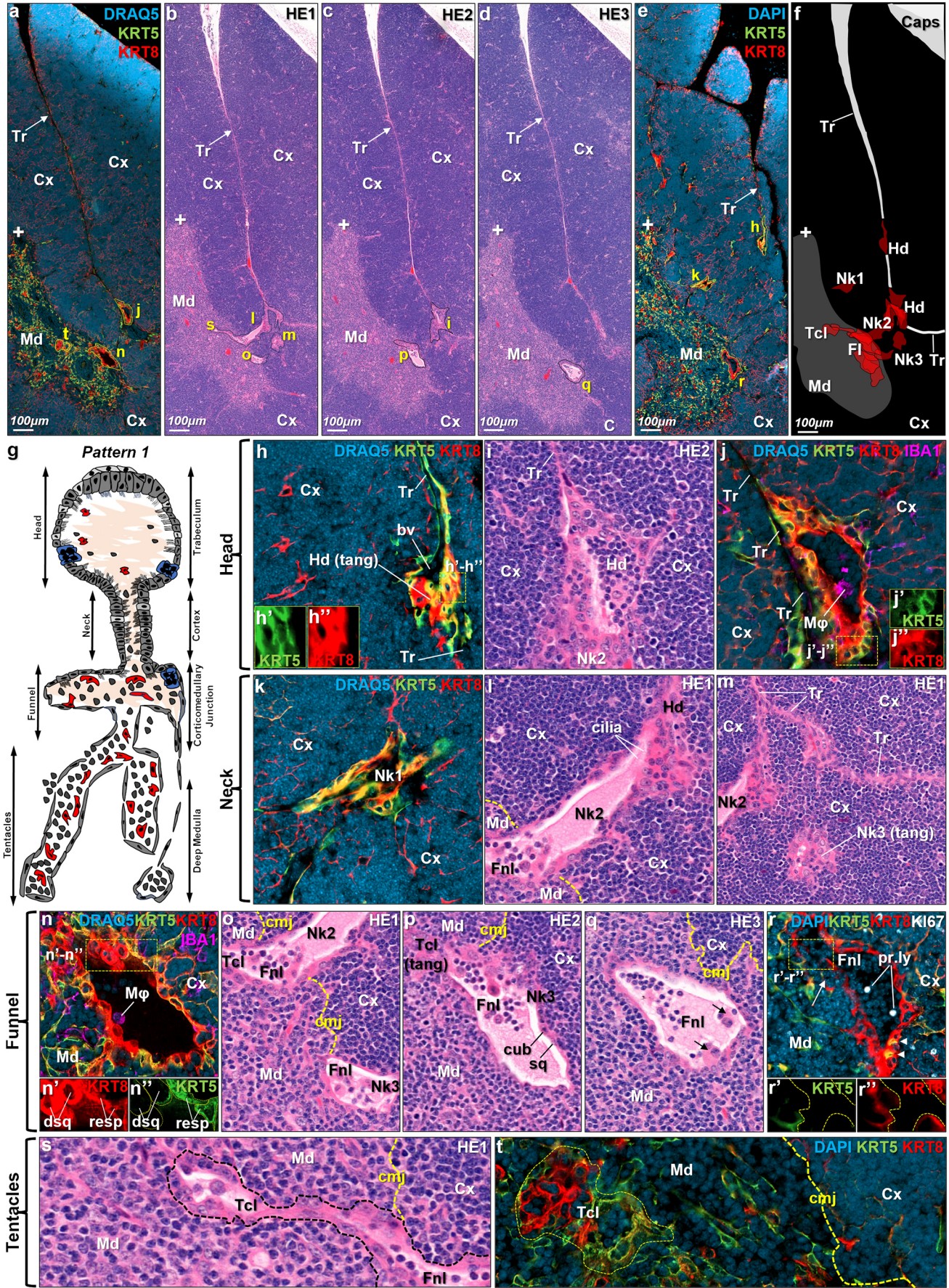

**Fig. 1.** See next page for legend.

**Fig. 1. Microanatomical organization of intrathymic epithelial networks.**
(A–E) Serial sections of murine thymus (5–25 μm apart) (control) revealing distinct regions of the intrathymic epithelial network, including structures resembling ducts, acini, alveoli and/or cysts, and their microanatomical localization within the thymic lobules. Sections are stained using immunofluorescence (A,E) or H&E (B–D). Tr, trabeculum; Cx, cortex; Md, medulla. The '+' symbol indicates corresponding microanatomical positions across sequential sections. Yellow letters refer to section show as magnified views in H–T. (F) Two-dimensional reconstruction of the composite intrathymic epithelial network (red overlays) in relation to thymic compartments. The network is divided into four portions: head (Hd; panels H–J), neck (Nk; K–M), funnel (Fnl; N–R) and tentacle (Tcl; S,T). For panels H–T, scale bars are omitted, as relative sizes can be inferred from the parental sections (A–E). (G) Schematic representation of the intrathymic epithelial network. Arrows on the left indicate anatomical portions, while arrows on the right denote predominant localization within thymic lobule. Lining epithelial cells are shown in gray, with luminal macrophages (red), thymocytes (dark gray), and desquamated cells (blue) occupying the cavities. The sizes and shapes of the different cell types are not drawn to scale, but represent a composite model of the knowledge obtained through the manuscript. (H–J) Magnified views of the cystic profiles in the head portion, as visualized in different planes using immunofluorescence (H,J) or H&E (I). A blood vessel (bv) is seen in the trabecular region adjacent to the head segment. IBA1$^+$ macrophages (Mφ, magenta) are present within the luminal cavity. (K–M) Magnified views of the cystic profiles in the neck portion, as visualized using immunofluorescence (K) and H&E (L,M). (N–R) Magnified views of the cystic profiles in the funnel portion, as captured via immunofluorescence (N,R) or H&E (O–Q). The corticomedullary junction (cmj) is outlined to demonstrate its spatial relationship to the funnel. Cavities are lined by respiratory epithelium (resp) and rarely contain proliferating lymphocytes (pr.ly) or desquamated cells (dsq). White arrow in R highlights a region in the medullary side of the funnel portion with KRT8 signal disruption. (S,T) Magnified views of the cystic profiles in the tentacle portion, as captured via immunofluorescence (S) or H&E (T). The cmj is annotated to illustrate the depth of tentacle extensions into the medulla. Images shown are representative of numbers of mice and sections as defined in the Materials and Methods.

(Davis and Wypych, 2021) and demonstrates a KRT5$^{hi}$ phenotype (Cheng et al., 2024; Halldorsson et al., 2007; Nagle et al., 1985). Indeed, we found a rare population of triangular/polygonal KRT8$^{lo}$KRT5$^+$ cells within the head portion of these epithelial networks, resembling a basal cell phenotype (Fig. S2H,I).

Interspersed among the KRT8$^{hi}$ population, there are also clusters of three to six interconnected KRT8$^+$KRT5$^+$ cells (Fig. 1H,J; Fig. S1G,I,L,N), possibly comprising parabasal (i.e. intermediate) cells transitioning from basal to other, terminally differentiated cell phenotypes.

As described above, the head portion of the epithelial network might begin as rostrally as near the thymic capsule and typically extends deep through the trabeculae, often circumnavigating multiple thymic lobules before invading into the cortex via its neck portion. This arrangement sometimes creates a unique feature at the rostral end of the head portion, where columnar cells reach their tallest height in the gland (Figs S1A–J, S2G–I). Owing to its pseudostratified structure, this region appears as a dense 'corona' composed of two or more cell layers enveloping the start of the epithelial network (Figs S1A–J, S2G–I). Although not always present, the corona cells often form direct cell-to-cell connections with the adjacent subcapsular/paratrabecular TECs (Figs S1G,L,L′,Q,Q′, S4K), usually on both sides of the trabeculum. The corona cells and their associated TEC continuum is not prominent in most epithelial networks in the involuted thymi (Fig. S3H). However, both corona cells and associated subcapsular/paraseptal TECs in normal thymi exhibit a KRT5$^+$KRT8$^+$ profile (Figs S1G,I,L,L′,N,N′,Q,Q′,S,S′). This profile is indicative of transitional stages for respiratory cells (Cheng et al., 2024; Davis and Wypych, 2021), but is an unusual feature for subcapsular/paratrabecular TECs, as KRT5 is generally associated with TECs localized to the corticomedullary junction and medulla (Colic et al., 1989; Farr and Braddy, 1989; Gupta et al., 2016; Kaneshima et al., 1987; Lee et al., 2011; Lomada et al., 2007; Loning et al., 1982; Popa et al., 2007). Indeed, the subcapsular localization of KRT5$^+$KRT8$^+$ TECs is a rare site where cortical TECs with KRT5 expression are reported to exist. The subcapsular KRT5$^+$KRT8$^+$ TECs that are physically associated with the corona typically reach up to between three and seven connected TECs, and exert various levels of KRT5 and KRT8 expression (Fig. S1G,I). Some subcapsular TECs in the corona–TEC continuum might predominantly express KRT5 with very low or absent KRT8 levels, although in most cases, the levels of KRT5 and KRT8 expression are equally high

**Table 1. Overview of intrathymic epithelial networks and their key distinguishing characteristics**

| Figure | Epithelial network portions | Unique or distinguishing features shown in network |
|---|---|---|
| Fig. 1 | Head-neck-funnel-tentacles (pattern 1) | • Archetypal epithelial network in control thymi, displaying all hallmark structural features described in this study |
| Fig. S1 | Head-neck (pattern 1) | • Prominent corona cell population within the head region<br>• Continuous interface between corona cells and subcapsular TECs |
| Fig. S2 | Head-neck-funnel (pattern 1) | • IF staining (DCLK1, AIRE, Ki67) reveals mTEC-epithelial network spatial relationships and proliferative capacity of lining cells<br>• Funnel lumens frequently contain thymocytes and macrophages<br>• Epithelial networks closely associate with blood vessels, often sharing a limiting membrane |
| Fig. S3 | Head-neck-funnel (pattern 1) | • Archetypal epithelial network retained in chemotherapy-involuted thymi, preserving key structural features<br>• Simplified architecture with reduced branching across network segments |
| Fig. S4 | Head-neck-funnel (pattern 1) | • Enrichment of corona cells in the head portion<br>• Continuous interface between corona cells and subcapsular TECs<br>• Dominance of ciliated cells along the network epithelium<br>• Mixed squamous and cuboidal epithelial morphology in the funnel portion |
| Fig. S5 | Tentacles (pattern 1) | • High variability in tentacle shape, epithelial lining and luminal content<br>• Tentacle portions interface with key intramedullary niches, including PVS and CMJ |
| Fig. S6 | Head-neck-funnel (pattern 2) | • Archetypal epithelial network in control thymi, encompassing a variant pattern, observed only in ~10% of the cases<br>• The variant head extends laterally within the cortex, lacks ciliation, and often forms a distended cisterna with infiltrating thymocytes and macrophages<br>• True medullary tentacles are absent, replaced by funnel-associated grape-like projections confined to the CMJ<br>• Neck portions are more elongated and branched, compensating for the altered trajectory of the head while maintaining classical epithelial features |

**Table 2. Summary of cell types, including mimetic cells, identified in each region of the epithelial network**

| Network portion | Lining/mimetic cell types | Luminal contents | | Abundance of cellular content |
|---|---|---|---|---|
| | | Non-cellular | Cellular | |
| Head | • Ciliated cells (including corona cells) <br> • Goblet cells <br> • Club cells <br> • Tuft cells <br> • Basal and parabasal cells <br> • Associated subcapsular/paraseptal tecs | Mildly eosinophilic, amorphous or flocculent secretory material | • Thymocytes <br> • Macrophages <br> • Debris | 25% of samples |
| Neck | • Ciliated cells <br> • Goblet cells <br> • Basal and parabasal cells <br> • Rare tuft-like cells | Amorphous/flocculent secretions | • Thymocytes <br> • Macrophages | Very rarely encountered |
| Funnel | • Ciliated cells (present, but less frequent) <br> • Goblet cells (rare) <br> • Secretory-columnar, club and club-like cells <br> • Squamous or flattened [microfold (M) cells, parabasal or intermediate cells, pulmonary ionocytes] | Amorphous/flocculent secretions | • Thymocytes (presumed SP thymocytes) <br> • Macrophages (subtype 1, large, eosinophilic, hypertrophic; subtype 2, small, starry-sky pattern, filled with apoptotic debris) (abundant) | Abundant |
| Tentacle | KRT8$^+$KRT5$^-$ squamous cells (identity not verified via electron microscopy) | Amorphous/flocculent secretions | • Thymocytes <br> • Macrophages (as in funnel) | Abundant, often entirely filling cavities |

This table presents a non-exhaustive list of epithelial cell types identified or suspected within distinct regions of the intrathymic epithelial networks. Additional cell types may be present; however, those lacking consistent morphological representation across samples were excluded to avoid overinterpretation.

(Fig. S1Q,Q′,S,S′). The precise nature of these subcapsular TECs remains uncertain. However, past studies have suggested that bipotent TEC progenitors, which co-express KRT5 and KRT8, might localize to the subcapsular zone. Although TEC progenitors are most commonly associated with the CMJ, the subcapsular zone might thus represent an alternative site where context-dependent factors bias differentiation toward the cTEC lineage instead of the mTEC lineage (Colic et al., 1989; Farr and Braddy, 1989; Gupta et al., 2016; Hirakawa et al., 2018; Kaneshima et al., 1987; Lee et al., 2011; Lomada et al., 2007; Loning et al., 1982; Nusser et al., 2022; Popa et al., 2007).

The neck portion of the epithelial network is often absent, because the trabeculae typically extend deep into the cortex, reaching very close to the CMJ, thus causing the head portion to transition directly into the funnel region. In most cases, however, one or more neck profiles branch out of the head to briefly traverse the underlying cortical region before reaching the CMJ (Fig. 1A–F,K–M,O,P; Figs S1P,R,T–Y, S2A–E,H–J,K–O, S4H,I). The single-neck pattern is especially observed in chemotherapy-involuted thymi, where such epithelial networks are almost always simplified, and consisting usually of one or rarely two neck portions directly linking the head to the funnel portion (Fig. S3A–G,H–J,O–Q). In normal thymi contrariwise, multiple neck profiles might branch out of a single head portion, often traveling vertically towards the CMJ and eventually converging into the same funnel portion (Fig. 1A–F,K–M,O,P), or distributed across different lobular segments (Fig. S2A–E,H–J,K–O). In all the above instances, the lining cells of the neck portions present with the lowest diversity and variability, but their morphology and cytokeratin expression profile highly resemble those of the head portion (Fig. 1K–M; Figs S1P,T–U,X,Y, S2H–O, S3H–J,P–Q).

Through detailed examination of the head and neck portions, we observed that cell proliferation is exceedingly rare among the lining cells, with Ki67$^+$ lining cells only sporadically detected (Fig. S2H, yellow arrow). Notably, Ki67$^+$ subcapsular TECs associated with corona cells were never identified. However, in most fields, a substantial number of Ki67$^+$ thymocytes were observed in the subcapsular and outer cortex regions, located in close proximity to epithelial networks (Fig. S2C,H,M). As expected, the lining cells of the head and neck regions also lacked expression of AIRE (Fig. S2A,F,K,P,U), which is typically expressed by mTECs (Anderson et al., 2002). Interestingly, however, a few scattered lining cells expressed the tuft cell marker DCLK1 (Fig. S2F,F′). DCLK1 is widely recognized as a lineage-defining marker for tuft cells, which comprise a thymic mimetic population instrumental in establishing self-tolerance against peripheral tuft cells (Bornstein et al., 2018; Miller et al., 2018). Peripheral tuft cells, though infrequent, are found interspersed within the gastrointestinal and respiratory epithelia, and are known for their role in eliciting immunological responses against luminal parasites (Abdel Wadood et al., 2025; Feng et al., 2024; Li et al., 2024).

The luminal content of the head and neck portions of epithelial networks is variable and comprises both non-cellular and cellular components. Hematoxylin and eosin (H&E) staining reveals a mildly eosinophilic, amorphous, or occasionally flocculent material within the lumen (Fig. 1L; Figs S1F,K,P,U, S2G,J,L,O, S3H–J,O–Q), likely secreted and/or modified by the lining cells. Occasionally, cellular debris is observed floating in the lumen or attached to the wall, suggesting a rapid turnover of the lining cells. Cellular components are predominantly concentrated in the lumen of the caudal regions of the epithelial networks, such as the funnel or tentacle structures, while the rostral (head and neck) regions are devoid of cellular content in the majority (~75%) of cases. In the remaining 25% of cases containing cells, these consist of a mixture of lymphoid and myeloid cells (Fig. 1I,J; Figs S1P,R,T, S2G). Lymphoid cells, likely thymocytes, appear to freely float in the luminal content and occasionally interact with the lining cells of the epithelial networks (Fig. 1I,J); macrophages touches the wall of the head, although their precise developmental stage remains unclear. A recent model on mimetic cells forming respiratory cysts implies that single-positive (SP) thymocytes could contact cystic cells for education and negative

selection by interacting with their presented self-antigens (Michelson and Mathis, 2022). However, the rostral position of the head and neck regions relative to the medullary regions raises questions about how SP thymocytes could reach this anatomical position. One hypothesis is that thymocytes that are present within the head and neck portion of epithelial networks might either represent negatively selected SP thymocytes migrating rostrally under the influence of the ciliated cells lining the network, or even represent earlier developmental stages infiltrating the epithelial networks directly from the cortical region for isolation/elimination or following an alternative intrathymic pathway of T cell maturation. Additionally, several macrophages are interspersed among the thymocytes in the rostral portions of the epithelial networks, as identified both by their characteristic morphology and the expression of the macrophage-specific marker IBA1 (also known as AIF1), which defines macrophage populations within the thymus (Fig. 1I,J).

### The funnel and tentacle portions of intrathymic epithelial networks

The central zone and caudal end of the epithelial network, comprising the funnel portion, are predominantly located at the CMJ (Fig. 1A–F; Figs S2A–E, S3A–G). This structure exhibits significant variation in size and shape, occasionally extending inward into the deep medulla through narrow alveolar projections resembling 'tentacles' (Fig. 1G). The funnel portion connects rostrally to the head and neck portions and manifests as a collection of broad channels extending laterally at the CMJ. This morphology gives this part its characteristic 'funnel' shape (Fig. 1G). Across all examined epithelial networks, including those in chemotherapy-involuted thymi, the funnel portion is consistently present, can be seen in longitudinal, oblique and cross-sectional views, and displays the highest structural and cellular diversity among all compartments.

The funnel portion establishes unique structural relationships with the surrounding thymic microenvironment. Owing to its lateral expansion along the CMJ, it is frequently associated with blood vessels and capillaries, potentially those involved in ETP homing in the thymic environment. Some of these blood vessels are in such close proximity to the funnel portion that they can appear to partially share the same connective tissues (Fig. S2P–R). Additionally, previous studies have reported that intrathymic cysts near the CMJ (here corresponding to the funnel portion of the epithelial networks) frequently exhibit discontinuous epithelium and basal lamina, allowing the adjacent thymocytes to infiltrate directly into the lumen (Khosla and Ovalle, 1986). Our observations align with these reports, as there are instances where the KRT5 and KRT8 signals are disrupted, particularly in the funnel side facing the medulla. As a result, medullary thymocytes can be seen to infiltrate the cystic lumen in such regions (Fig. 1R–R″, white arrow).

The principal cells lining the funnel portion exhibit diverse morphologies, ranging from tall-columnar to flattened-squamous cells, with cuboidal morphologies representing an intermediate state (Fig. 1N–R, Figs S2P–Y, S3J–L,R–T, S4L–P). Although frequent exceptions exist, the funnel portion is often polarized, with columnar and cuboidal segments predominantly positioned at the cortical side, whereas flattened-squamous cells align with the medullary side (Fig. 1P, Fig. S4P). Moreover, cellular composition is highly heterogeneous. Although H&E staining and KRT5 and KRT8 immunofluorescence provide an overview of their diversity, these methods lack the resolution essentially to distinguish the individual cell types. Nevertheless, certain epithelial subtypes can be clearly identified, including ciliated cells and goblet cells (which are rarer in comparison). Groups of non-ciliated and

non-mucinous secretory-columnar cells are very prevalent (Fig. 1O–Q; Figs S2K–Y, S3S,T, S4L–O) and are expected to correspond to club, tuft and/or neuroendocrine cells (Davis and Wypych, 2021). As discussed above, a frequent epithelial subpopulation within the funnel lining exhibits a flattened-squamous morphology (Fig. 1N–R; Fig. S2Q–T,V–Y), which, at first glance, resembles cells from the distant respiratory epithelium, that is, alveolar type I and II. However, our electron microscopy investigations (shown below) strongly suggest that such flattened-squamous cells correspond to microfold (M) cells, a specialized subset of cells that regulates immune responses against luminal antigens, previously reported in most mucosal epithelia (Kimura, 2018).

Unlike the head and neck regions of the epithelial networks, the funnel portion rarely exhibits KRT5$^+$KRT8$^-$ lining cells (Fig. 1N,R–R″; Fig. S2R,S,W,X), suggesting a possible lack of a basal cell population, similar to what is seen in the rostral portion of the network. Instead, a subset of epithelial cells lining the funnel portion can exhibit KRT5$^+$ basolateral surfaces, but KRT5 expression is generally mild and the apical surfaces either completely lack KRT5 expression or express it at a minimal level (Fig. S4P–P″). Double-positive KRT5$^+$KRT8$^+$ lining cells are also rare, but when present, they appear in small clusters of two to five cells, most often positioned on the cortical-facing side rather than the medullary-facing side (Fig. 1R; Fig. S2S,X, white arrowheads). The lining cells of the funnel portion do not express AIRE or DCLK1, clearly suggesting they are not mTEC$^{hi}$ or tuft cell aggregates (Fig. S2P,U). Taken together, these findings suggest that the funnel portion of the epithelial networks contains a limited number of progenitor and intermediate cell populations, raising the possibility that lining cell renewal occurs rostrally, with newly developed epithelial cells 'sliding' along the axis of the epithelial network. Alternatively, the progenitor cells of the funnel portion might possess a distinct phenotype and lineage marker composition, differing from those found in the rostral segments.

Although not always present, a varying number of tubular prolongations occasionally extend from the funnel portion, traversing deeper into the medullary parenchyma to form the so-called 'tentacles' (Fig. 1G,S,T; Fig. S5A–P). In cross-sections, most tentacles appear as small circular profiles (Fig. S5A–P), whereas longitudinal views clearly depict their origins within the funnel (Fig. 1A,B,S,T). However, in some cases, sequential sectioning reveals no apparent connection between the tentacles and the funnel portion of the epithelial network. This observation raises two possibilities – either tentacles are independent cystic structures unrelated to the epithelial network, or they are indeed connected but collapse in certain regions, creating the impression of isolated cavities (Fig. 1G).

Tentacles are distributed at various depths within the medullary parenchyma, following no distinct patterns in their trajectory. They frequently border or entirely envelop the perivascular spaces (PVS) of the medulla, which are epithelium-free (KRT5$^-$) regions, and can be filled with SP thymocytes that have successfully undergone negative selection before emigrating from the thymus. This positioning raises the intriguing possibility that tentacles, as the most caudal or distal extensions of the epithelial network, might load negatively selected SP thymocytes, offering an alternative intrathymic egress pathway distinct from conventional blood vessel-mediated exit routes within the PVS. Supporting this hypothesis, tentacles are often thin-walled and primarily lined by KRT8$^+$KRT5$^-$ squamous cells, with KRT5$^+$ mTECs forming well-demarcated nest-like structures around them (Fig. S5F–F″,H–H″,J–J″, L–L″,N–N″,P–P″). This suggests that thymocytes migrating from the medulla to the tentacle lumen must

first navigate through and interact with a broad meshwork of mTEC subsets, before entering the tentacle lumen. However, this working model should be interpreted with caution, as our studies do not offer any mechanistic and dynamic experiments to prove an alternative thymocyte emigration pathway.

The luminal content of the distal portion of epithelial networks – that is, the funnel and tentacles – closely resembles the rostral segments, which contain abundant thymocytes and myeloid cells, predominantly macrophages, all freely floating within an eosinophilic, amorphous and occasionally flocculent material (Fig. 1O–Q,S; Figs S2O,Q,T,V,Y, S4L–O, S5E,G,I,K,M,O). However, a key distinction is that the funnel and tentacles are densely populated with cellular components, with few (if any) regions completely devoid of cells (Fig. 1P,Q; Fig. S5G). The only consistent exception occurs in chemotherapy-involuted thymi, where epithelial networks completely lack cellular content (Fig. S3I–K), likely due to delayed immunological recovery and impaired ETP homing. The cellular component is mostly composed of thymocytes, although abundant myeloid cells are interspersed among them, the latter identified primarily as macrophages, based on their high IBA1 expression (Fig. 1N; Fig. S2S,S′,X, S4P, S5F). Two macrophage subtypes are distinguished – a highly eosinophilic, hypertrophic oval-shaped macrophage with abundant cytoplasm (Fig. 1Q, black arrows), and a smaller macrophage, exhibiting a 'starry sky' pattern and various contents, most notably apoptotic debris (Figs S2T,V,Y, S4M,N, S5E). Thymocytes within the funnel lumen are predominantly non-proliferative, supporting the premise that they represent a non-expanding SP thymocyte subset. However, rare Ki67$^+$ profiles were observed within this region (Fig. 1R, Fig. S2R,W), raising an alternative possibility that rather than being exclusively post-selection SP thymocytes, some luminal thymocytes might have entered the cystic lumens at an earlier developmental stage. In this scenario, these thymocytes would retain their proliferative capacity and potentially undergo the final stages of thymic selection inside the epithelial network, interacting with respiratory-specific or other peripheral antigens present in this microenvironment.

Small clusters of KRT8$^{hi}$ epithelial cells are occasionally found within the luminal content, most often attached to other lining cells of the cavity (Fig. 1N–N″). Although the identity of these KRT8$^{hi}$ cells remains uncertain, certain investigators (Khosla and Ovalle, 1986) have speculated that they might represent apoptotic cells that have detached from the luminal wall. However, if these cells are indeed indicative of the rapid turnover of the epithelial lining, three key observations would be expected: (1) they would appear more frequently and broadly throughout epithelial networks, (2) they would be captured across all compartments rather than being almost exclusively restricted to the funnel portion, and (3) they would likely exist as individual detached cells rather than as tightly connected clusters, given that cellular junctions are among the first structures to dissolve during apoptosis in outermost epithelial layers. Regardless of the above, so-called 'hillock cells', descendants of the basal cell lineage with high cycling capacity and high KRT13 expression, have been reported as segments of continuous stratified squamous epithelium mixed within the pseudostratified epithelial cells (Deprez et al., 2020; Montoro et al., 2018). Given that the clusters appear tightly connected, resemble a nest-like structure corresponding to a stratified layer and are always attached to the epithelial network wall rather than floating freely (Fig. 1N–N″), it is plausible they represent a subset of hillock cells. However, their rarity, consistent with limited documentation of hillock cells, makes definitive conclusions difficult at this stage. Further studies, including lineage-tracing approaches, are necessary to better characterize these epithelial cell clusters.

## Intrathymic epithelial network variants

A detailed investigation of over 30 epithelial networks in the adult murine thymus revealed that ∼90% follow the reported pattern (Fig. 1G). However, as previously noted, variations occasionally occur. These include the absence or extreme narrowing of the neck portion, the presence of multiple head or neck portions connecting caudally to the same funnel, and the absence of tentacles or their apparent isolation due to the collapse of their rostral connection to the funnel. Although one or more of these structural variations might be simultaneously present in each network, imparting a unique signature to each, the overall organization, comprising the head, neck, funnel and tentacles, along with their distinct cellular and luminal contents, remains highly conserved. However, ∼10% of the observed epithelial networks exhibit a structurally divergent pattern, following a head–neck–funnel organization without tentacles (Fig. S6A–J). Although these networks retain the head–neck–funnel structure, they present key structural deviations from the archetypal form.

The most striking difference is an elongated head portion, which, rather than crossing the trabecular connective tissue to reach the CMJ, instead travels laterally within the paratrabecular cortex, avoiding the trabecular connective tissue entirely (Fig. S6K–P). Unlike in conventional epithelial cysts, ciliated cells are often absent from the head portion of this variant. Instead, this head variant forms a distended or dilated cisterna, which is predominantly lined by flattened squamous KRT5$^-$KRT8$^+$ or KRT5$^+$KRT8$^+$ cells in its trabeculum-facing side and by KRT5$^-$KRT8$^+$ squamous or cuboidal cells in its outer cortex-facing side (Fig. S6K–P). Additionally, this head variant contains a significant number of thymocytes and macrophages (Fig. S6K–P), in stark contrast to the conventional epithelial networks, where the rostral regions, particularly the head portion, rarely contain such cellular infiltrates (see above).

Similar to conventional epithelial networks, these variants also feature multiple rather than single neck portions branching from the caudal head, running through the outer, mid and deep cortex to reach the CMJ, where they expand into the funnel (Fig. S6Q–V). However, because the head portion extends laterally along the paratrabecular cortex and not vertically toward the trabecular tip and the CMJ, the neck portions tend to be more elongated, covering tall cortical segments before reaching the CMJ (Fig. S6Q–V). Despite these structural differences, the lining cells of the neck portions remain consistent with those found in the conventional epithelial networks. Likewise, the neck portions of these variants give rise to a funnel portion that laterally traverses the CMJ, maintaining a similar composition in both epithelial lining and luminal content compared to the conventional epithelial networks (Fig. S6W–AB). As in conventional epithelial networks, the funnel of the variant network also exerts discontinuities in its epithelial wall (Fig. S6AB, white arrows), possibly resulting in the infiltration of surrounding thymocytes.

Unlike conventional epithelial networks, these variants do not develop true tentacles that extend deep into the medulla (Fig. S6J). Instead, the caudal portion of the funnel gives rise to tentacle-like projections, resembling grape-like clusters or slightly elongated tubular structures (Fig. S6Y,AA,AB). These projections remain integrally connected to the funnel, functioning as part of its expanded architecture rather than forming separate distal branches within the deep medullary parenchyma. Despite these key morphological differences, the epithelial lining of the caudal portion largely remains consistent with the conventional pattern, suggesting that the primary structural deviations occur mostly within the rostral portions.

### General ultrastructure of intrathymic epithelial networks

As described above, immunofluorescence and light microscopy reveal a progressive shift in the cytological organization of intrathymic epithelial networks from a secretory-dominant rostral region (head and neck) to attenuated squamous epithelia in the caudal region (tentacles), with a central funnel zone exhibiting complex, mixed morphologies. However, identifying the specific cell types lining these pseudostratified epithelia is challenging without lineage-specific markers. Moreover, these structures, situated in the unique thymic microenvironment, might adopt respiratory-like but distinct morphologies and lineages. To further resolve this complexity, we employed transmission electron microscopy (TEM), which confirmed regional differences, with a focus on the highly variable neck–funnel region at the CMJ (Fig. 2). The funnel typically presents a round-to-oval cavity between cortex and medulla, although its convoluted path can cause partial or tangential views of the lining cells (Fig. 2). The narrower neck traverses the cortex (Fig. 2). TEM findings confirmed the light microscopy results, showing amorphous granular material and suspended cells within the neck–funnel lumen (Figs 1,2; Figs S1–S6). The funnel portion is frequently located near

capillaries and post-capillary venules of CMJ vasculature (Fig. 2), the main entry site for thymocyte progenitors. Thymocytes at various developmental stages are abundant around the neck and funnel areas (Fig. 2), along with a variety of cTEC and mTEC subsets (Fig. 2, white asterisks). Importantly, the epithelial lining of the neck–funnel portion shows striking variability in cell size, shape and electron density. Most of the cavity is lined by a single epithelial cell layer in direct contact with the thymic parenchyma, often lacking a basal lamina (Fig. 2). In some areas, a more structured epithelial layer where fragments of basal lamina are observed (see Fig. 4E). Despite not necessarily touching a basal lamina, the lining cells display apicobasal polarity, with apical surfaces projecting into the lumen, and range morphologically from cuboidal to columnar to squamous (Fig. 2), highly consistent with immunofluorescence findings.

### Ultrastructural features of epithelial cells lining the epithelial networks

Based on the data presented thus far (Figs 1,2) and also supported by prior studies (Khosla and Ovalle, 1986), the intrathymic cavities appear to have a respiratory character, specifically that the epithelial

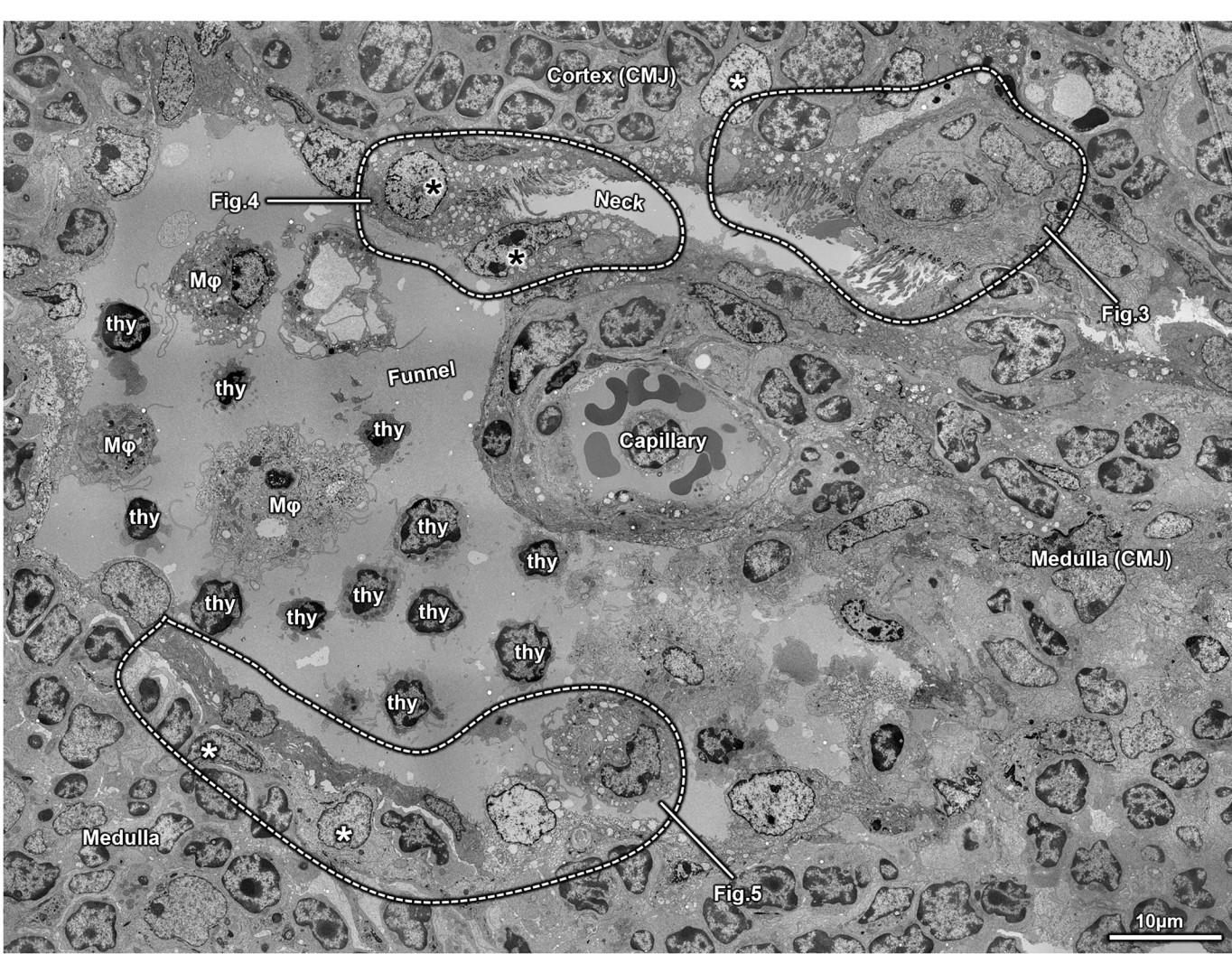

**Fig. 2. General ultrastructure of the neck-funnel portion of an intrathymic epithelial network.** The funnel and neck components are directly connected, although this connection is partially obscured by two tangentially sectioned lining cells (black asterisks). The funnel resides in the cortico-medullary junction (CMJ), whereas the neck extends rostrally toward the cortex. A capillary is observed in close apposition to the cortical side of the funnel. Both the neck and funnel exhibit marked heterogeneity in their lining epithelial cells. Selected lining cells within the white-dotted enclosures are shown at higher resolution in Figs 3, 4 and 5. A few scattered thymic epithelial cells (TECs; white asterisks) are associated with the cyst-lining epithelium. The cyst lumen contains primarily thymocytes (thy) and macrophages (Mφ). Images shown are representative of at least N=5 repeats.

cells lining these cavities resemble those of the upper respiratory system, such as bronchial or bronchiolar pseudostratified epithelia. However, if these epithelial networks are not simply cystic remnants of the third pharyngeal pouch during thymus organogenesis as originally supposed (Dooley et al., 2005a,b), but actually represent organized and/or mimetic cell formations (Michelson and Mathis, 2022), the lining cells would be expected to undergo intrathymic adaptations and therefore contextually differ from traditional respiratory mucosal epithelia. To explore this hypothesis, we characterized the epithelial cells lining intrathymic cavities using TEM, categorizing them according to the latest classification scheme, as proposed by Davis and Wypych (Davis and Wypych, 2021). These categories include basal cells (including both basal and parabasal or intermediate), ciliated cells, club cells, goblet cells, tuft cells, microfold cells, pulmonary neuroendocrine cells, pulmonary ionocytes and hillock cells (Davis and Wypych, 2021). Below, we provide a detailed description of all epithelial cells identified in the neck and funnel portions (Figs 3–5).

### Ciliated cells

Ciliated cells are tall, columnar epithelial cells essential for clearing microorganisms, mucus and debris from the respiratory system via coordinated, rhythmic beating of their apical cilia (Davis and Wypych, 2021). Within murine intrathymic epithelial networks, we observed a higher prevalence of ciliated cells in the rostral head and neck regions (Fig. 1), as confirmed by electron microscopy (Fig. 2). Prior studies have described two transcriptional states in proximal respiratory ciliated cells (Travaglini et al., 2020) and two ultrastructural variants, electron-pale and electron-dark (hereafter, just called pale and dark), distinguished by ground substance and cytoplasmic features (Khosla and Ovalle, 1986). Consistent with this, we identified both pale and dark ciliated cells, predominantly lining the neck region (Figs 3A–C, 4A–D). Pale ciliated cells (Figs 3B, 4B–D) are large, with euchromatic, round or irregular-shaped nuclei, and finely dispersed heterochromatin. Their cytoplasm is rich with organelles, including a compact Golgi, clustered mitochondria, free ribosomes, polyribosomes and numerous vesicles. Smooth endoplasmic reticulum (sER) profiles with fine granular content are occasionally observed (Figs 3B, 4C), although these less developed than in secretory or lipid-processing cells. Dark ciliated cells (Fig. 3C) are similarly large but exhibit denser cytoplasm with swollen mitochondria lacking distinct cristae near the basal bodies of cilia and an extensive tubulovesicular network (Fig. 3C). As expected, the hallmark feature of all ciliated cells, both pale and dark, is the presence of multiple apical cilia with the characteristic 9+2 axonemal pattern (Figs 3B,C, 4D). Overall the differences between pale and dark ciliated cells are consistent with prior observations (Khosla and Ovalle, 1986), but it remains to be seen whether these reflect to the alternate transcriptional states documented by molecular studies (Travaglini et al., 2020).

### Club and club-like cells

Club and club-like cells are columnar-shaped secretory cells characterized by an elongated, euchromatic nucleus with an indented nuclear membrane and prominent nucleoli (Fig. 4A,B). They possess a well-developed biosynthetic apparatus, supporting their many roles in secretion, detoxification and epithelial regeneration (Hong et al., 2001; Tata et al., 2013; Zuo et al., 2018). These cells are non-ciliated (Fig. 4E) and typically form interdigitating lateral membranes with adjacent cells, reinforced by junctional complexes (i.e. desmosomes) (Fig. 4F). Their most distinctive cytoplasmic organelle is the highly abundant, irregularly shaped sER, which forms

a tubulovesicular network (Fig. 4E), possibly participating in xenobiotic metabolism and lipid biosynthesis. This network often contains electron-lucent or amorphous material (Fig. 4E), likely representing surfactant precursors or detoxification byproducts. Mitochondria are mildly elongated ovals and rarely exhibit lamellar cristae. Instead, the mitochondria frequently appear swollen and are often interspersed among sER profiles (Fig. 4E,F). Correspondingly, autophagic vacuoles with electron-lucent or flocculent material tend to accumulate in these cells (Fig. 4E,F). Unlike other cells lining the epithelial network, these cells sometimes contain prominent cytoskeleton filaments, forming evident bundles (Fig. 4E, black asterisk). Compared to mature club cells, these intrathymic club and club-like cells do not exhibit abundant electron-dense granules with myelin figures, which is the typical profile of surfactant proteins. This suggests that intrathymic club cells might either function simply as intermediates in the differentiation toward other respiratory cells or serve an alternative role unrelated to surfactant production in the thymus. Indeed, recent classification schemes of respiratory epithelia suggest that club and club-like cells might serve as progenitors for goblet cells and ciliated cells (Davis and Wypych, 2021).

### Parabasal or intermediate cells

Parabasal and intermediate cells are randomly interspersed among well-differentiated subsets of respiratory epithelial cells. These poorly differentiated cells are generally small and characterized by a narrow rim of cytoplasm surrounding an oval- or bean-shaped nucleus (Figs 5A,B). The nucleus is highly euchromatic, with finely dispersed heterochromatin (Fig. 5E–G). Occasionally, the perinuclear cisternae appear slightly dilated (Fig. 5E–G, pn-cis, black asterisks), indicating continuity with the rough endoplasmic reticulum (rER). The cytoplasm contains scant organelles, reflecting their immature state compared to well-differentiated respiratory epithelial cells. Notably, they exhibit few mitochondria, as well as free ribosomes, and a limited number of cytoplasmic vacuoles and transport vesicles filled with electron-lucent material (Fig. 5E–G). Another morphological feature of parabasal or intermediate cells is their cuboidal or triangular shape, with a distinct cellular extension protruding into the lumen (Fig. 5A,B). This luminal exposure serves as a key distinguishing feature, separating them from basal cells, which share a cuboidal or triangular morphology and reflect the earliest luminal progenitors, but are instead situated at the base of the pseudostratified epithelium, tightly squeezed between adjacent cells and lacking direct luminal contact (Davis and Wypych, 2021). As such, parabasal or intermediate cells reflect a transcriptionally distinct subpopulation of basal cells, that has been committed to luminal differentiation and described as transit-amplifying cells (Hajj et al., 2007; Hong et al., 2004; Rock et al., 2009, 2010).

### Pulmonary ionocytes

Pulmonary ionocytes are frequently observed in small clusters. These large cells feature an elongated, eccentrically located moderately electron-dense nucleus filled with euchromatin and peripheral clumps of heterochromatin (Fig. 5A,C). The cytoplasm is dominated by mitochondria and an extensive membranous system (Fig. 5C). Mitochondria are small, oval, spindle-shaped or occasionally branched, with characteristic tubulovesicular cristae and electron-dense matrix. Pulmonary ionocytes possess two distinct membranous systems. The first is a well-developed, flattened system of membranes, typically located on the apical side facing the lumen. This system, previously described as a modified sER (Khosla and Ovalle, 1986), is not definitively linked to this origin. The flattened membrane profiles are interconnected by

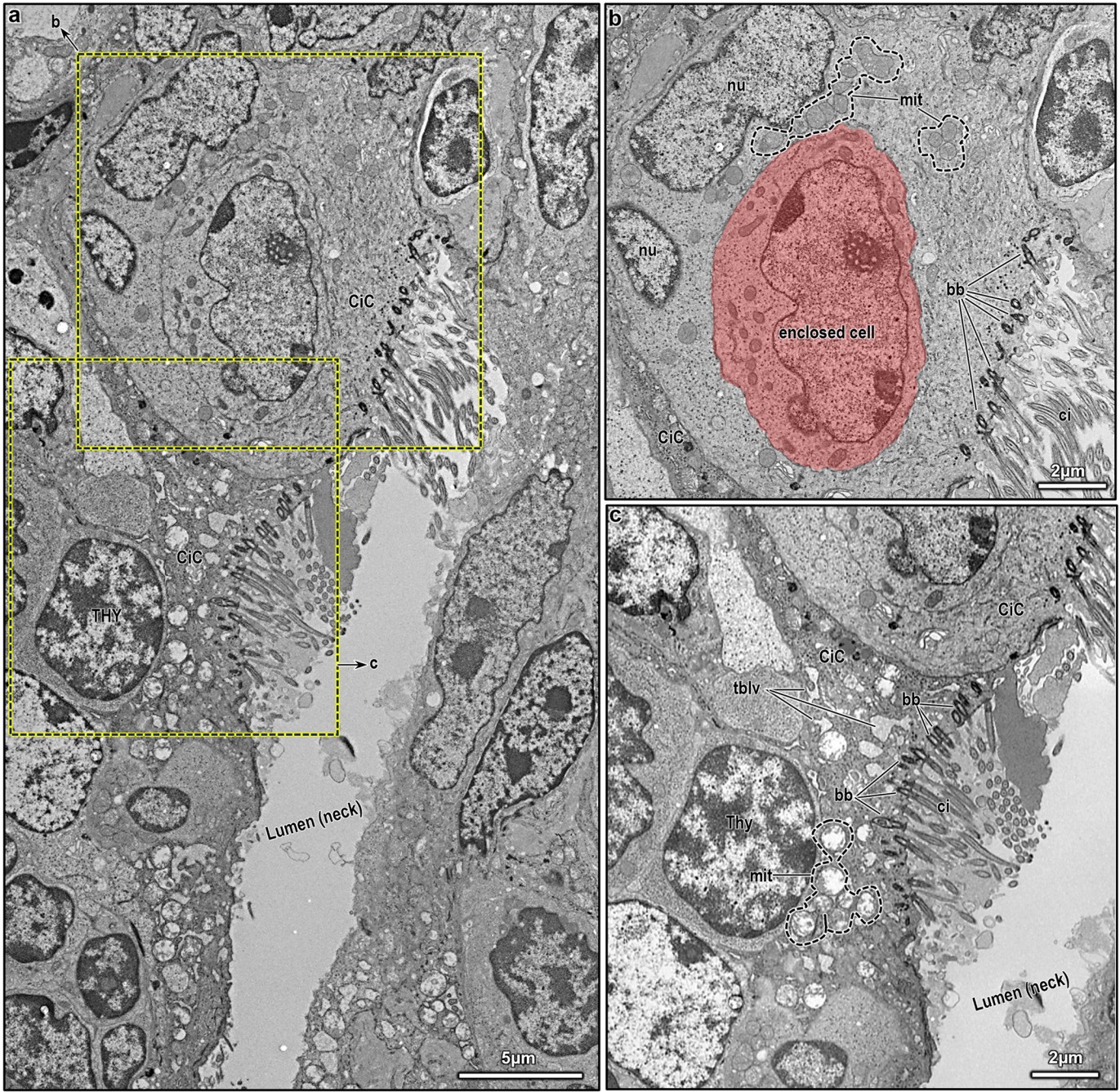

**Fig. 3. Ultrastructure of ciliated cells in the neck portion of an intrathymic epithelial network.** (A) Two adjacent ciliated cells (CiC) lining the neck portion are shown, each with multiple cilia extending into the lumen. The upper cell is a pale ciliated cell with an indented nucleus, whereas the lower is a dark ciliated cell, whose nucleus is not visible in this plane of section. (B) Higher magnification of the pale ciliated cell (marked area from A). The cell displays clustered mitochondria (mit) and a distinctly indented, bilobed nucleus (nu). Enclosed within the ciliated cell is another cell featuring a euchromatic, oval-shaped nucleus with a prominent nucleolus with nucleolemma pattern (red overlay). Multiple basal bodies (bb) are polarized toward the lumen, anchoring motile cilia (ci). A portion of the neighboring dark ciliated cell is visible in the bottom left. (C) Higher magnification of the dark ciliated cell (marked area from A). This cell exhibits features suggestive of elevated biosynthetic activity, including abundant, swollen mitochondria (mit) and a prominent tubulovesicular network (tblv) occupying much of the cytoplasm. As in B, basal bodies (bb) are polarized toward the luminal surface, anchoring motile cilia (ci). A portion of the neighboring pale ciliated cell is visible at the top. Images shown are representative of at least N=5 repeats.

narrow strips (Fig. 5J), indicating that they form a continuous sac rather than independent vacuoles or intracellular cysts. The second membranous system comprises a moderately developed rER, appearing as relatively short profiles (Fig. 5J). These profiles often wrap around mitochondria (Fig. 5I,I′), although they are also interspersed between the flattened membranous system. Pulmonary ionocytes exhibit a few microvilli but abundant irregular infoldings

and protrusions. Finally, multiple dense, double-membraned bodies are present, often with irregular shapes, including circular, tubular, bean-shaped, or crescent-like forms (Fig. 5I,J).

**Microfold cells**

Microfold cells are specialized squamous-shaped epithelial cells, oriented horizontally in relation to the lumen of the cavity

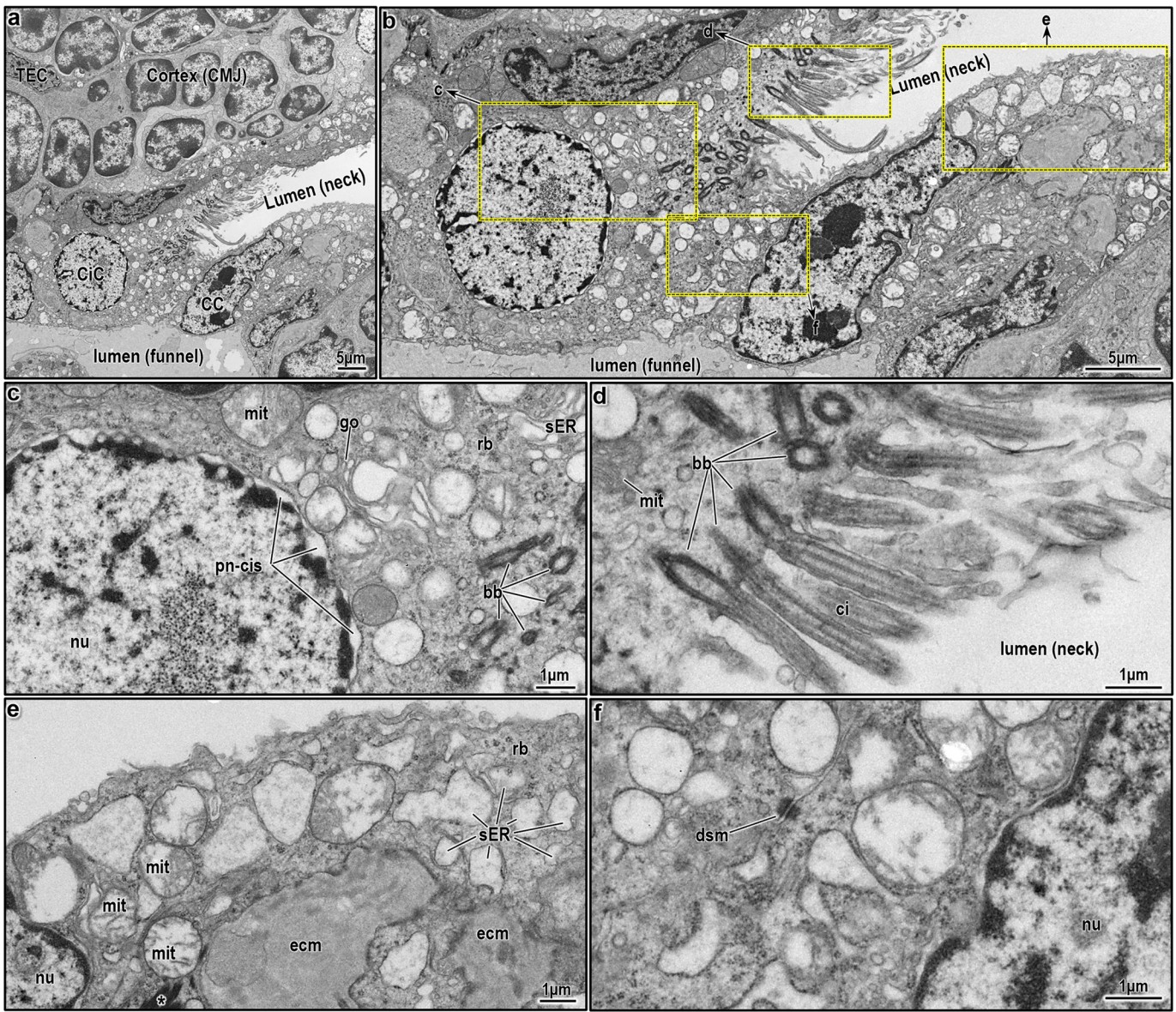

**Fig. 4. Ultrastructure of lining epithelial cells at the neck–funnel nexus of an intrathymic epithelial network.** (A) Two epithelial cells lining the neck-funnel nexus are sectioned tangentially, creating the misleading appearance of an obstructed passage between the neck and funnel lumens. The left cell is a pale ciliated cell (CiC), and the right one is reminiscent of a club cell (CC). (B) Higher magnification of the cells shown in A, highlighting their distinct morphologies. (C) Ultrastructure of the pale ciliated cell, showing abundant mitochondria (mit), a well-developed Golgi (go), free ribosomes (rb) and a saccular network comprising sER. The nucleus (nu) is predominantly euchromatic with marginal chromatin condensation, whereas the perinuclear cisternae (pn-cis) appear swollen in regions. (D) Apical part of the pale ciliated cell, revealing numerous basal bodies (bb) and longitudinally sectioned motile cilia (ci) projecting into the neck lumen. The nearby mitochondria occasionally display tubulovesicular structures in place of fully lamellar cristae. (E) Ultrastructural features of the club cell, which is anchored to extracellular matrix (ecm) without a clearly defined basal lamina. The cell contains swollen mitochondria (mit), free ribosomes (rb), and extensive tubulovesicular network of sER saccules. (F) Tangential sectioning of the ciliated and club cells (from A and B) obscures the direct continuity between neck and funnel lumens, which would be visible in a different imaging plane. A desmosomal junction (dsm) connects the two cells, and the club cell nucleus is seen on the right. Images shown are representative of at least *N*=5 repeats.

(Fig. 5A,D). Overall, they exhibit electron-dark ground substance in both their nucleus and their cytoplasm when compared to other well-differentiated cells lining the intrathymic epithelial networks (Fig. 5A,D). Their nuclei are small and irregularly shaped with deep indentations, but otherwise mostly euchromatic (Fig. 5D). Their cytoplasm contains particularly elongated profiles of rER, small clusters of free ribosomes, small Golgi complexes and numerous mitochondria (Fig. 5K,L). Unlike typical secretory cells, microfold cells possess lysosomes and only a few apical secretory granules (Fig. 5K,L). The hallmark feature of microfold cells is their

extensive basolateral infoldings (Fig. 5K,L), accompanied by the absence of significant apical specializations, aside from a few short microvilli. These basal infoldings do not adhere to any basal lamina, thus creating direct connections between the extracellular matrix (Fig. 5L, black asterisks) and the cavities formed between the infoldings (Fig. 5K,L, white asterisks). When viewed in longitudinal sections, the basolateral infoldings appear serpentine and remarkably long, significantly increasing the surface area of the basolateral plasma membrane (Fig 5K,L, white asterisks). This additional surface potentially creates the characteristic 'pocket' of

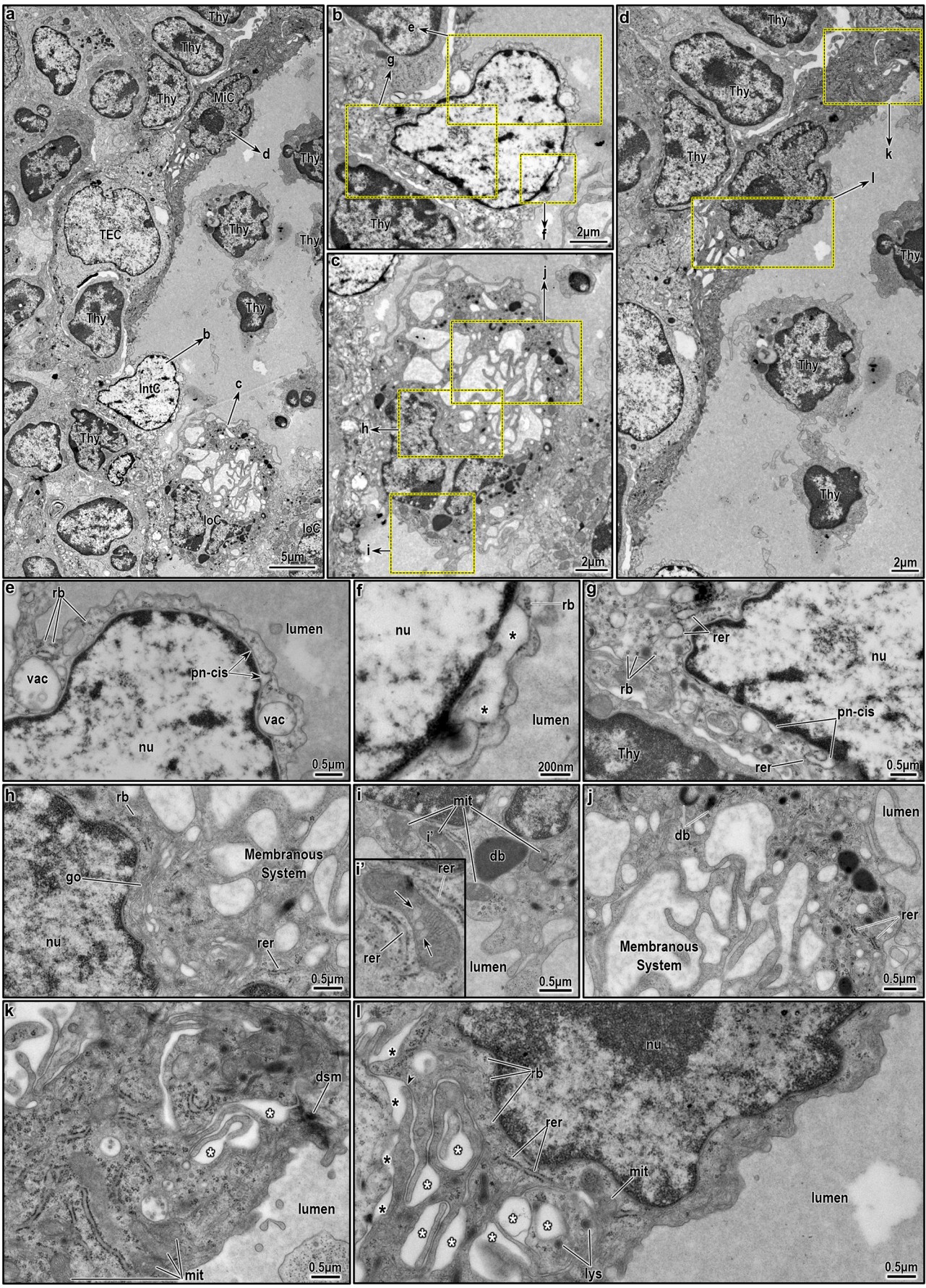

**Fig. 5.** See next page for legend.

**Fig. 5. Ultrastructure of lining epithelial cells in the funnel portion of an intrathymic epithelial network.** (A) Three adjacent epithelial cells lining the funnel portion are shown sequentially along the thymic parenchyma (left side of the image) – a squamous-shaped microfold cell (MiC), a parabasal or intermediate cell (IntC) and an ionocyte (IoC). Floating thymocytes are visible in the lumen. (B–D) Higher magnification of the cells as in A, highlighting their distinct morphologies; the intermediate cell features a triangular euchromatic nucleus and scant cytoplasm protruding into the lumen (B); the ionocyte displays its characteristic membranous system (C); and the microfold cell appears flattened and elongated along the luminal axis (D). (E–G) Magnified views of the marked areas in B, revealing ultrastructural features of the intermediate cell. The cytoplasm contains sparse organelles, including free ribosomes (rb), elongated rER (rer) and a few vacuoles (vac). Perinuclear cisternae (pn-cis) are swollen in certain regions and show direct continuity with the rER (e). A close-up view highlights the bloated perinuclear cisternae (black asterisks). (H–J) Magnified views of the marked areas in C, detailing the ultrastructure of the ionocyte. The cell exhibits a prominent membranous system composed of elongated sacs and a well-developed biosynthetic apparatus, including Golgi complexes (go), rER (rer) and ribosomes (rb) positioned near the nucleus. Dense bodies (db) (I) and mitochondria with tubulovesicular morphology (I′) are also present. The membranous system is extensively developed, with large and small sacs interspersed and surrounded by dense bodies and rER (J). (K,L) Magnified views of the marked areas in D, illustrating the ultrastructure of the microfold cell. The cell has squamous morphology and basal pocket forming multiple infoldings (white asterisks), which directly connect to the basal surface (black arrowhead, black asterisks). This pocket typically harbors an immune cell, but in this instance, it appears unoccupied. Images shown are representative of at least N=5 repeats.

the microfold cells when another cell migrates into it. In tissues such as the intestine, these pockets are almost always occupied by dendritic cells or lymphocytes, thus enabling the microfold cells to perform transcytosis and transferring of luminal antigens to the professional antigen-presenting cells (Del Castillo and Lo, 2024; Kimura, 2018; Mabbott et al., 2013). Under such conditions, they possibly never assume a purely squamous shape. In our investigations, however, the observed microfold cells exhibited unoccupied basal pockets (Fig. 5A). This finding contrasts with the results of an elegant study form the 1980s (Khosla and Ovalle, 1986) who also reported such squamous cells lining the intrathymic cavities surrounding large lymphocytes. In that study, the authors observed that thymocytes frequently occupied these 'pockets' and were entirely encased by the cytoplasm of squamous cells, with intact membranes for both the thymocytes and the lining epithelial cells, suggesting a direct yet non-invasive interaction (Khosla and Ovalle, 1986). Notably, at the time that this 1986 study was conducted, it was not widely recognized that microfold cells also exist in respiratory epithelia, likely due to rarity (Davis and Wypych, 2021) and, as such, the authors provided a description but never assigned a clear identity to squamous cells.

## Ultrastructural morphology of solitary ciliated cells in the thymic medulla

Both historical and modern TEC classification systems in rodents and other species have described mTECs featuring large intracellular cystic structures lined with microvilli and cilia (Ito and Hoshino, 1966; Nabarra et al., 2001; Sugimoto et al., 1977). In our study, we identified a distinct population of isolated ciliated mTECs, characterized by a large, eccentrically positioned intracellular lumen juxtaposed against the nucleus (Fig. 6A). These cells are typically found near the cortico-medullary junction (CMJ) and outer medulla, and remain separate from the organized intrathymic epithelial networks, described above (Figs 1–5). Although sometimes solitary, they more frequently appear in small clusters, which are either homogeneous (multiple interconnected ciliated mTECs) or

heterogeneous, coexisting with various mTEC subsets (Fig. 6A). Owing to their location near the CMJ, these cells often interface with large thymocytes, that is thymocytes with abundant cytoplasm more likely to correspond to double-negative (DN) than double-positive (DP) or single-positive (SP) thymocytes (Fig. 6A).

The ultrastructural features of these solitary ciliated cells include euchromatic nuclei, and they often have abundant cytoplasm, prominent mitochondria, a well-developed Golgi, abundant ribosomes and polyribosomes, and occasionally an enlarged tubulovesicular network (Figs. 6B–D), all of which are hallmarks of active biosynthesis that might support the early processes of T cell development. Although some have proposed a role for ciliated mTECs as respiratory-like mimetic populations involved in negative selection (Michelson and Mathis, 2022), our findings are also indicative of a different functional context. Their frequent contact with DN as opposed to SP thymocytes raises questions about this proposed role and warrants further investigation.

The defining feature of ciliated mTECs is a prominent intracellular lumen (5–10 µm in diameter) with variable contents, ranging from electron-lucent or granular material to electron-dense granules, some of which resemble viral particles, although their identity remains uncertain (Fig. 6E,F). The lumen is lined with microvilli and numerous cilia projecting inward in multiple orientations. Basal bodies are distributed around the periphery and base of the lumen, producing both longitudinal and cross-sectional ciliary profiles (Fig. 6E,H,I). These cilia display the typical 9+2 axoneme (Fig. 6F), a hallmark of motile epithelial cilia. To our knowledge, this inward orientation of cilia within an intracellular lumen is a unique feature not reported in other mammalian epithelial cell types besides in ciliated mTECs.

In chemotherapy-involuted thymi, ciliated mTECs are readily identified due to reduced thymocyte density and often appear in larger clusters near the CMJ and outer medulla (Fig. 6G,H). However, neighboring mTECs often show signs of cytotoxic damage, complicating identification. Interestingly, decreased thymocyte numbers enhance visualization of surviving mTECs and their desmosomal connections (Fig. 6I). Despite the cytotoxic stress, these cells do not exhibit obvious ultrastructural abnormalities. However, the small sample size limits our ability to assess potential chemotherapy-induced changes in cilia density, basal body integrity or cellular morphology. Future studies are needed to explore these questions in greater depth.

The unusual inward orientation of cilia in these cells might reflect a lack of apicobasal polarity within the unique architecture of the thymic microenvironment. Unlike ciliated epithelial cells in polarized tissues, such as the respiratory tract or oviduct, ciliated mTECs lack basal lamina and tight junctions, key features that typically define apical specialization. In the absence of these structural cues, ciliary components might reorganize inward toward an intracellular lumen, representing an adaptation to their non-polarized epithelial context. Although single-section analyses provide valuable insight, they are inherently limited in capturing the full complexity of the ciliated lumens. To overcome this, we employed high-resolution 3D osmium-thiocarbohydrazide-osmium scanning electron microscopy (OTO-SEM) array tomography (Figs. 7A–C; Movie 1). The intracellular ciliated lumen occupies a substantial volume within the mTEC (Fig. 7A,B; Movie 1). Segmented reconstructions at different planes allowed the layered visualization of the spatial relationship between the lumen, plasma membrane and nucleus of the cell (Fig. 7A–C). Basal bodies were distributed circumferentially along the luminal surface, without evidence of polarized organization (Fig. 7A–C; Movie 1). Multiple

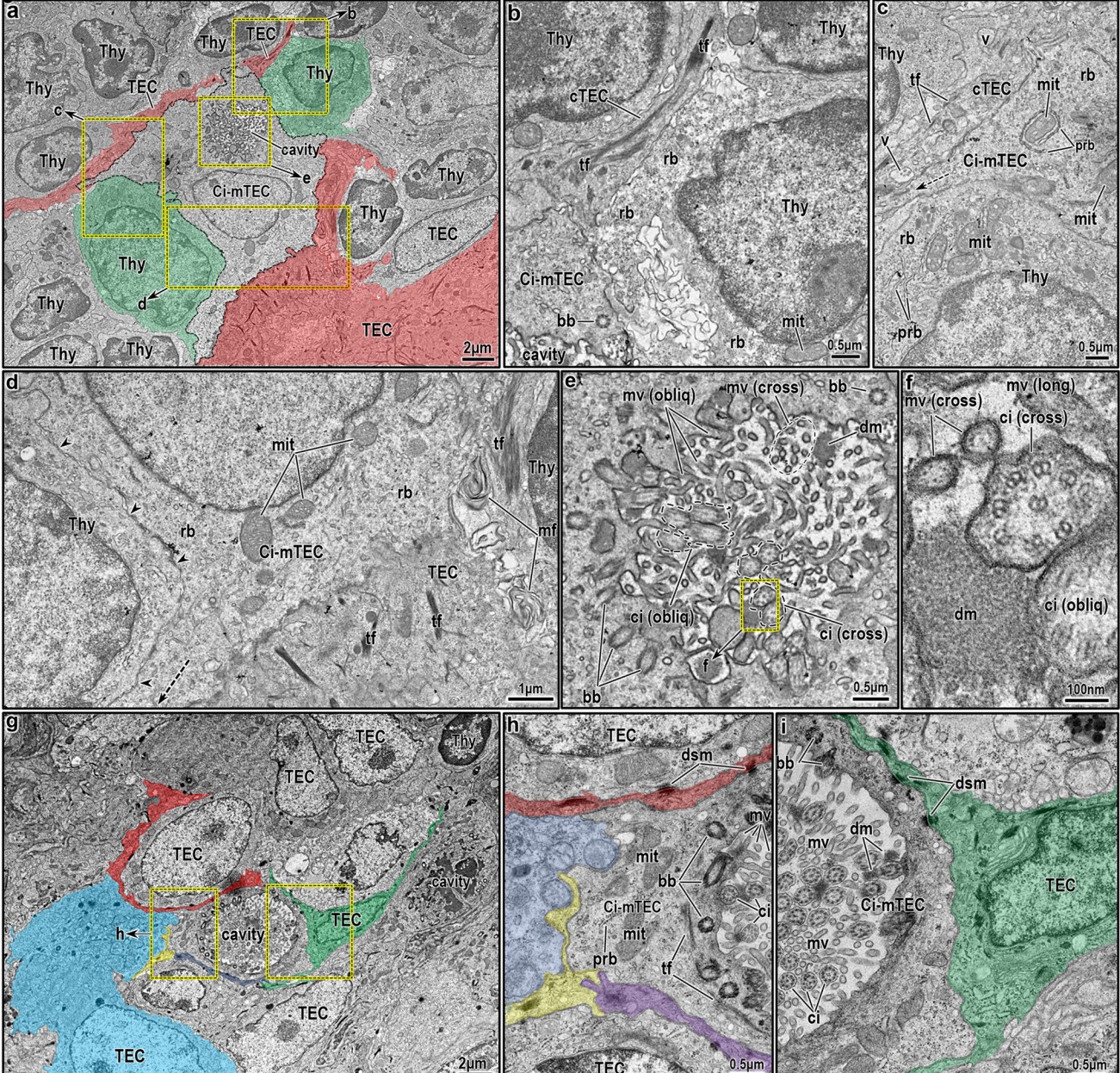

**Fig. 6. Ultrastructure of isolated ciliated mTECs in the normal and involuted murine thymus.** (A) Overview of the corticomedullary junction in a normal murine thymus, showing a centrally located isolated ciliated mTEC (Ci-mTEC) surrounded by large thymocytes (green overlays), and by projections of other TECs extending from both the cortical (upper) and medullary (lower) sides (red overlays). These TEC extensions make direct contact with the surface of the ciliated cell. (B–E) Magnified views of the marked areas in A, highlighting ultrastructural details of various compartments of the Ci-mTEC. (B) Interaction between the Ci-mTEC (lower left corner) and a cortical TEC (cTEC) projection, which contains prominent tonofilaments (tf). The cell body of the cTEC is not visualized in this section and is likely located in the mid- or deep cortex. A large thymocyte, likely at an early progenitor stage given its abundant cytoplasm and its pre-Golgi complex development, is also present. (C) Additional interactions with cTEC projections are shown, featuring tonofilaments (tf) and vacuoles (v). Organelles of the Ci-mTEC are also visible, including ribosomes (rb) and mitochondria (mit) closely associated with polyribosomes and/or elongated rER profiles. A large DN1-stage lymphocyte, characterized by abundant cytoplasm and organelles, is in contact with the Ci-mTEC. (D) At the bottom right corner, the Ci-mTEC borders an mTEC, likely part of a Hassall's corpuscle, as suggested by the presence of tonofilaments (tf), degenerative material, and myelin figures (mf). Arrowheads indicate contact with a large thymocyte, likely at an early progenitor stage, while the dotted arrow points to cytoplasmic projections of the Ci-mTEC. (E) Ultrastructural view of the cystic cavity within the Ci-mTEC, showing an abundance of microvilli and cilia projecting into the lumen in multiple orientations (cross-sectional, longitudinal and oblique). Dense material (dm) is also observed within the lumen. (F) Magnified view of the marked areas in E, detailing cilia (ci), microvilli (mv) and dense material (dm) within the cavity. (G–I) Overview of the corticomedullary junction in a partially restored thymus, 14 days post-chemotherapy-induced involution. Two cavities belonging to isolated Ci-mTECs are visible, now surrounded predominantly by various TECs. Owing to involution, thymocytes are largely absent, and most TECs appear collapsed and have lost distinct membrane features (e.g. cytoplasmic projections), making it difficult to classify them as cTEC or mTEC subtypes. (H,I) Magnified views of the marked areas in G, showing fine cellular interactions, including desmosomal junctions (dsm) between the Ci-mTEC and multiple neighboring TECs, each indicated with different overlays (red, cyan, yellow, green and magenta). These interactions are likely enabled by the structural collapse of surrounding TECs and absence of thymocytes. Notably, the cilia and microvilli of the Ci-mTEC cavities remain unaffected by chemotherapy or involution and are clearly visible. Multiple basal bodies (bb) are oriented circumferentially around the cystic cavity. Images shown are representative of at least N=5 repeats.

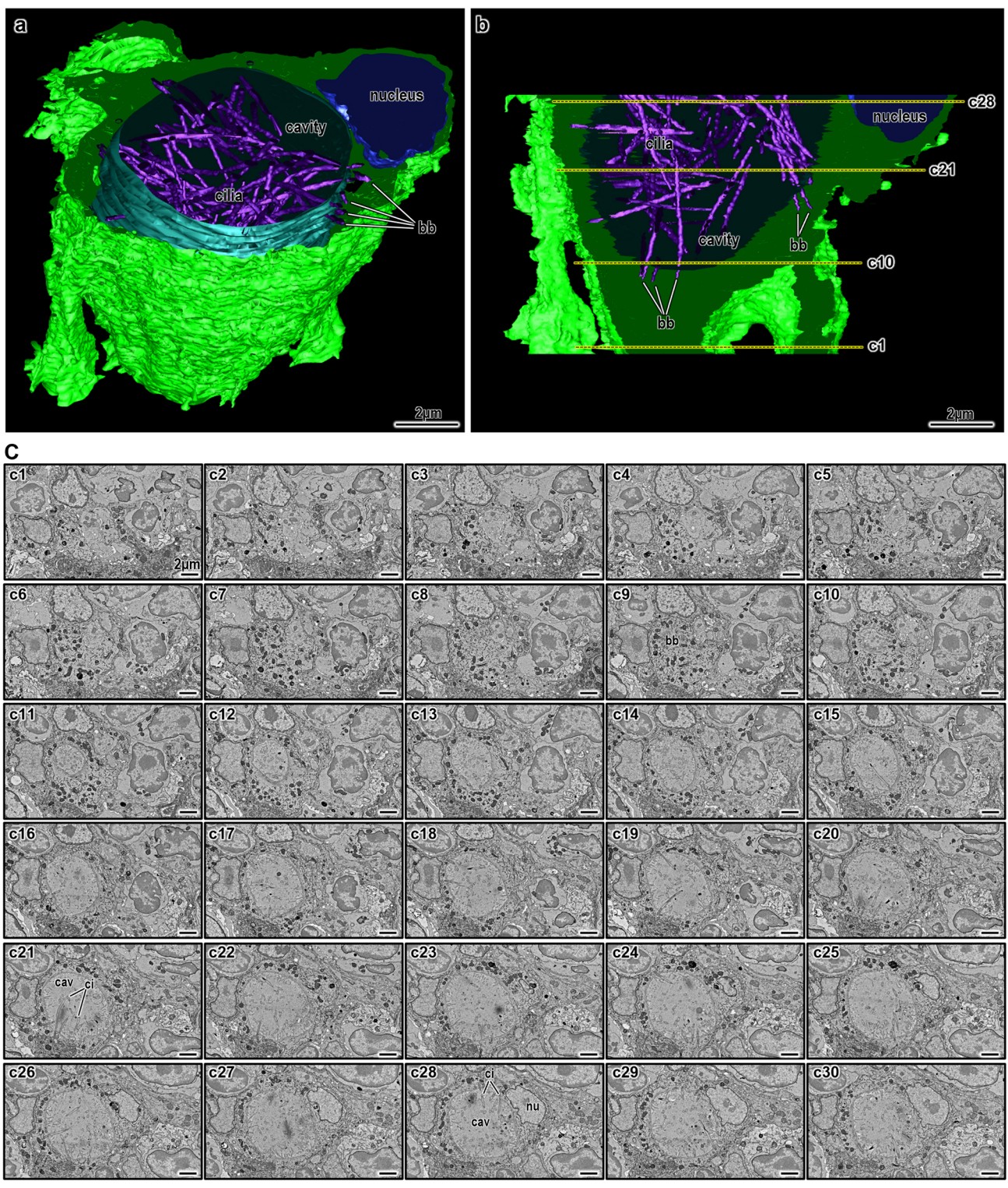

**Fig. 7. Three-dimensional reconstruction of an isolated ciliated mTEC in the murine thymus, at day 14 post-chemotherapy.** (A,B) 3D reconstruction of the ciliated cell. Oblique view (A) and sagittal slice (B) displaying the lateral profile and internal architecture. The outer cellular membrane is shown in green, the nucleus in blue, the inner luminal wall in cyan, and the cilia in magenta. The yellow dashed lines (B) indicate the approximate positions of transverse sections c1, c10, c21 and c28. (C) Selected consecutive transverse sections (c1 through c30), picked out as every fifth section of a total of 151 sections used for the 3D reconstruction shown in A and B, obtained via the OTO-SEM method. Structural components include basal bodies (bb), luminal cavity (cavity), cilia (ci), mitochondria (mit) and nucleus (nu). Images shown are representative of at least *N*=3 repeats.

cilia projected from different directions into the lumen, where they intermingled freely in complete absence of coordinated orientation (Fig. 7A–C; Movie 1).

## Conclusions

This study presents two major findings that expand our understanding of thymic epithelial architecture. First, we provide

the most detailed characterization to date of intrathymic epithelial networks, revealing that structures historically described as isolated cysts, ducts, or glands are, in fact, interconnected epithelial networks traversing from the capsule to the deep medulla. Through light microscopy, immunofluorescence and TEM, we define conserved organizational zones – the head, neck, funnel and tentacles – each with distinct histological features, and epithelial cell types resembling those of the respiratory lineage, including ciliated, club, goblet, microfold and ionocyte-like cells, among others.

Second, we identify and reconstruct in three dimensions a novel epithelial phenotype – the isolated ciliated mTECs, which are not part of the epithelial networks. These cells possess large intracellular lumens lined with inward-facing cilia, which are arranged without apicobasal polarity and distributed circumferentially around the lumen. To our knowledge, this is the first ultrastructural and volumetric analysis of such a configuration in any mammalian epithelial cell. Their function remains unknown; they might represent an adaptive morphological response to the non-polarized thymic environment, a reprogrammed epithelial state or a vestigial remnant of organogenesis.

We also report that both organized intrathymic epithelial networks and isolated ciliated mTECs are present not only in normal thymi but also in those involuted by chemotherapy. This indicates that these cells and/or structures are resilient and conserved following cytotoxic damage and acute thymic involution. Whether their persistence reflects a role in endogenous regeneration or simply intrinsic resistance to injury remains to be determined.

While characterizing the epithelial networks, we found no expression of AIRE in any lining cells across all compartments, indicating they do not represent the mTEC^hi thymic epithelial subset (Yano et al., 2008). Interestingly however, some cells expressed the tuft cell marker DCLK1 (Miller et al., 2018), suggesting that the individual lining cells might correspond to mimetic subsets. However, our analysis did not encompass a broad panel of TEC and mimetic markers. As the developmental trajectories and evolutionary origins of thymic mimetic cells continues to expand (Nusser et al., 2025), future studies should aim to comprehensively characterize the epithelial lining using an expanded set of markers, such as Ly51, β5t and MHC-II (Kadouri et al., 2020).

A fundamental limitation of ultrastructural and histomorphological studies, including the present work, is their reliance on morphological and phenotypic resemblance for cell identification, rather than definitive lineage-tracing or molecular marker validation. Although TEM provides unparalleled resolution of cellular architecture, it does not reveal lineage or function. As such, classification depends on expert interpretation and might be influenced by factors like sectioning orientation, transitional states or incomplete visualization of defining features. To mitigate these sources of bias, all identifications were made through consensus by a multidisciplinary team with expertise in thymic biology, ultrastructural biology and veterinary pathology. Even so, we advise readers to interpret classifications with appropriate caution and recommend that future studies combine ultrastructural imaging with molecular or genetic approaches to improve accuracy and reproducibility, including interobserver agreement statistics (McHugh, 2012).

Altogether, these discoveries raise compelling questions about epithelial plasticity, unconventional modes of thymocyte interaction and the developmental origin of these structures. By establishing a structural and cellular framework for both intrathymic epithelial networks and isolated ciliated mTECs, this work provides a foundation for future studies exploring their molecular identity, lineage potential, immunological relevance and importance to regenerative processes.

## MATERIALS AND METHODS

### Mouse model of cyclophosphamide-induced thymic involution
All studies involving mice were conducted in accordance with NIH regulations concerning the care and use of experimental animals with the approval of IACUC of Molecular Imaging, Inc. (Ann Harbor, MI), a facility accredited by the Association for Assessment and Accreditation of Laboratory Animal Care (AAALAC), or with the approval of Albert Einstein College of Medicine Animal Care and Use Committee. Because the thymus is sexually dimorphic (Hince et al., 2008; Hun et al., 2020; Taves and Ashwell, 2022), male and female populations should not be pooled together, as there might be major differences. However, the focus of the current study was not on determining microanatomical differences between male and female mice, and as such, only female mice have been used. All mice used in these studies are of FVB/N background (NIH). Cyclophosphamide monohydrate (Thermo Scientific) was reconstituted at a concentration of 25 mg/ml in sterile phosphate-buffered-saline (PBS). Each mouse in the experimental group received an intraperitoneal (i.p.) dose of 200 mg/kg of body weight of cyclophosphamide monohydrate (Ctx; Acros Organics) in sterile PBS (200 µl total volume), every 3 days, for a total of three doses, beginning at the age of 6–7 weeks. The control or vehicle-treated (Ctrl) mouse group instead received an i.p. injection of 200 µl sterile PBS. The last day of treatments was designated as 'day 0'. The mice were then categorized into seven subgroups by a researcher who was not aware of the experimental treatment. For light microscopy studies, mice from either Ctrl or Ctx subgroups were killed using cervical dislocation on days 3 (Ctrl, $N$=13; Ctx, $N$=18), 7 (Ctrl, $N$=14; Ctx, $N$=15), 10 (Ctrl, $N$=13; Ctx, $N$=16), 14 (Ctrl, $N$=14; Ctx, $N$=16), 21 (Ctrl, $N$=12; Ctx, $N$=22), 28 (Ctrl, $N$=14; Ctx, $N$=20) and 35 (Ctrl, $N$=5; Ctx, $N$=12). For electron microscopy studies, mice from either Ctrl or Ctx subgroups were killed using cervical dislocation on days 3 (Ctrl, $N$=1; Ctx, $N$=3), 7 (Ctx, $N$=1), 10 (Ctx, $N$=1), 14 (Ctrl, $N$=1; Ctx, $N$=3), 21 (Ctrl, $N$=1; Ctx, $N$=1) and 28 (Ctx, $N$=1). All samples were processed for TEM, except for one Ctx-treated mouse at post-Ctx day 14, which was processed via osmium-thiocarbohydrazide-osmium (OTO), which is suitable for SEM 3D-array tomography.

### Histology
Thymi from Ctrl and Ctx-treated mice were fixed in 10% neutral buffered formalin for 48 h, processed and paraffin embedded as described previously (Karagiannis et al., 2017). To identify cysts and intrathymic epithelial networks, which are relatively rare within the thymic parenchyma, we accrued sequential 5-µm-thick sections in 25-µm-deep intervals for up to 250 µm of total depth into the thymic tissue. These 5 µm sections were deparaffinized and dehydrated in double changes of xylene and 100%, 95%, 70% ethanol, respectively, followed by rehydration. Four to seven sections in this case were selected for H&E staining, and blanks were also kept in-between for immunofluorescence staining (see below). The H&E staining was conducted as previously described (Karagiannis et al, 2017); paraffin sections were deparaffinized in xylene and rehydrated through a graded ethanol series (100%, 95%, 70%) into water, stained with hematoxylin for 5 min and differentiated for 5 s, counterstained with Eosin Y for 1 min 45 s, then dehydrated through graded ethanols (70%, 95%, 100%), cleared in xylene, and mounted with Permount (SP15-100, Thermo Fisher Scientific). All H&E-stained slides were carefully examined for the identification of intrathymic epithelial networks. Of the 204 animals examined, we identified intrathymic epithelial networks in 33 animals (Ctrl, $N$=17; Ctx, $N$=16) with each timepoint represented by at least two mice.

### Immunofluorescence
Several deparaffinized thymic sections, preferentially between H&E sections, were used for immunostaining of lineage-specific markers. For antigen retrieval, slides were embedded in citrate (Novus Biologicals) and EDTA (Epredia) in a steamer for 21 min. The blocking step was performed with 5% goat serum (R&D systems, cat. no. S13110) in PBS with 0.05% Tween 20 for 1 h, after which the slides were incubated with the following primary antibodies overnight at 4°C. Antibodies used were against: KRT5 (chicken IgY, purified polyclonal; cat. no. 905904, BioLegend, 1:300), KRT8 (guinea pig, polyclonal; cat. no. BP5075, Origene, 1:100), AIRE (rat, monoclonal; cat. no. 14-5934-82, Invitrogen, 1:500); DCLK1 (rabbit,

monoclonal; cat. no. MA532657, Thermo Fisher Scientific, 1:100), IBA1 (rabbit, polyclonal; cat. no. 013-27691, Wako, 1:100), KI67 (rabbit, polyclonal; cat. no. 12202S, Cell Signaling, 1:500). The secondary antibody incubation was performed with the antibodies at 1:200 dilution for 50 min [goat anti-rabbit-IgG conjugated to Alexa Fluor 488 (cat. no. A11034, Invitrogen), goat anti-rabbit-IgG conjugated to Alexa Fluor 647 (cat. no. PIA32733, Invitrogen), goat anti-chicken-IgY conjugated to Alexa Fluor 488 (cat. no. A11039, Invitrogen), goat-anti-rat-IgG conjugated to Alexa Fluor 647 (cat. no. A21247, Invitrogen), goat anti-guinea pig-IgG conjugated to Alexa Fluor 555 (cat. no. A21435, Invitrogen)], followed by nuclear staining for 5 min with DAPI (Novus Biologicals) or DRAQ5 fluorescent probe (Abcam), and mounting with ProLong Gold anti-fade mounting reagent (Invitrogen). Slides were left to dry overnight at dark and subjected to digital scanning via a 3D HISTECH P250 Flash III automated slide scanner, using a 20×0.75NA objective lens into a multichannel overlay image, as described in prior work from our group (Borriello et al., 2020; Coste et al., 2020; Karagiannis et al., 2017; Sharma et al., 2021). All slides used were subjected to the same color adjustment for each marker in CaseViewer v2.4 software.

## Electron microscopy

Fresh thymic tissues were processed either for TEM ($n$=8), or with an OTO method ($n$=1), for 3D-SEM array tomography. For TEM, the samples were fixed with 2.5% glutaraldehyde and 2% paraformaldehyde in 0.1 M sodium cacodylate buffer, postfixed with 1% osmium tetroxide followed by 2% uranyl acetate, dehydrated through a graded series of ethanol and embedded in LX112 resin (LADD Research Industries, Burlington VT). Ultrathin sections were cut on a Leica Ultracut UC7, stained with uranyl acetate followed by lead citrate and viewed on a JEOL 1400 Plus transmission electron microscope at 120 kV. For OTO-SEM, the samples were immersion fixed in 2.0% paraformaldehyde and 2.5% glutaraldehyde in 0.1 M sodium cacodylate buffer, then processed using a modified National Center for Microscopy and Imaging method of OTO (doi:10.17504/protocols.io.36wgq7je5vk5/v2). In brief, samples were post fixed with reduced osmium, treated with thiocarbohydrazide, further stained with osmium, en bloc stained with uranyl acetate, further stained with lead aspartate, dehydrated in a graded series of ethanol and embedded into LX112 resin. 55-nm-thick sections were cut on a Leica Artos microtome using a Diatome AT 35° knife and picked up on freshly glow-discharged silicon wafers. Sections were observed on Zeiss Supra 40 field emission scanning electron microscope in backscatter mode, using an acceleration voltage of 8.0 kV. Regions of interest were collected with ATLAS 5.0, using a pixel size of 6.0×6.0 nm and a dwell time of 6.0 µs. Images were aligned and segmentation was done using the IMOD suite of programs (Kremer et al., 1996; Mastronarde, 1997). Briefly, stacks of images were aligned in IMOD using Midas and cells of interest were manually segmented out using the drawing tools in 3DMOD. Ciliated TECs were identified by large lumens filled with cilia and microvilli.

## Acknowledgements

Several investigators would like to acknowledge the following support for shared instrumentation: the Zeiss Supra 40 Field Emission Scanning Electron Microscope (1S10RR025554-01A1), JEOL 1400Plus Transmission Electron Microscope (1S10OD016214-01A1) and the 3DHISTECH P250 High-Capacity Slide Scanner (1S10OD026852-01A1) in the Analytical Imaging Facility (AIF), both acquired through Shared Instrumentation Grants from The National Institutes of Health (NIH). The AIF is also partially funded by the NCI Cancer Center Support grant (P30CA013330).

## Competing interests

The authors declare no competing or financial interests.

## Author contributions

Conceptualization: G.S.K.; Data curation: V.D.; Formal analysis: S.V., S.D., H.G., G.S.K.; Funding acquisition: G.S.K.; Investigation: L.G.-C., F.M., G.S.K.; Methodology: S.V., L.G.-C., X.N., J.C., R.A., V.D., F.M., G.S.K.; Project administration: V.D., F.M., G.S.K.; Resources: L.G.-C., M.K.L., H.G., R.A., V.D., F.M.; Software: L.G.-C., V.D., F.M.; Supervision: G.S.K.; Visualization: S.V., G.S.K.; Writing – original draft: S.V., G.S.K.; Writing – review & editing: L.G.-C., V.D., F.M., G.S.K.

## Funding

This work was supported by new investigator start-up funds, provided to G.S.K. from the Montefiore-Einstein Comprehensive Cancer Center (National Cancer Institute, P30CA013330), and the Young Investigator Award, awarded to G.S.K. by the Rally Foundation. Open Access funding provided by Albert Einstein College of Medicine. Deposited in PMC for immediate release.

## Data and resource availability

All relevant data and details of resources can be found within the article and its supplementary information.

## First Person

This article has an associated First Person interview with the first author of the paper.

## Peer review history

The peer review history is available online at https://journals.biologists.com/jcs/lookup/doi/10.1242/jcs.264079.reviewer-comments.pdf

## Special Issue

This article is part of the Special Issue 'Cilia and Flagella: from Basic Biology to Disease', guest edited by Pleasantin Mill and Lotte Pedersen. See related articles at https://journals.biologists.com/jcs/issue/138/20.

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
