## [Peer Review File · Journal of Cell Science]

Beyond cysts - organization of epithelial networks in the murine thymus

Stepan Vodopyanov, Leslie Gunther-Cummins, Sophia DesMarais, Xheni Nishku, Joseph Churaman, Hillary Guzik, Rotem Alon, Vera DesMarais, Frank Macaluso and George S. Karagiannis

DOI: 10.1242/jcs.264079

Editor: Pleasantine Mill

Review timeline

Original submission:	12 April 2025
Editorial decision:	17 June 2025
First revision received:	27 June 2025
Accepted:	11 August 2025

Original submission

First decision letter

MS ID#: jcs.264079

MS TITLE: Structural Organization of Respiratory Elements in the Murine Thymus

AUTHORS: Stepan Vodopyanov; Maria Lagou; Leslie Gunther-Cummins; Sophia DesMarais; Xheni Nishku; Joseph Churaman; Rotem Alon; Hillary Guzik; Vera DesMarais; Frank Macaluso; George Karagiannis

ARTICLE TYPE: Research Article

Dear George and team,

We have now reached a decision on the above manuscript.

To see the reviewers' reports and a copy of this decision letter, please go to:

As you will see, the reviewers gave favourable reports but raised some points that will require amendments to your manuscript. I hope that you will be able to carry these out because I would like to be able to accept your paper for our Special Issue on Cilia and Flagella, depending on further comments from reviewers.

Apologies for the the lengthy review time for this manuscript. It was challenging to secure reviewers and chasing reviews. Hopefully, it will be worth the wait. As you can see all three reviewers and this editor feel that this is an important and innovative work and are keen to see it revised. No further experiments are being asked for but rather more discussion, more information on key points of genetic background, age, and sex of mice used in the study and strong textual edits to simplify the narrative as directed by Reviewer 3 to increase accessibility and impact. It would also be worth discussing the recently published work from the Boehm lab and how it relates to your study- Nusser, A., Thomas, O.S., Zhang, G. et al. Developmental trajectory and evolutionary origin of thymic mimetic cells. Nature (2025). <https://doi.org/10.1038/s41586-025-09148-y>.

Reviewer 1

This paper is reporting unexpected, quite fascinating results. While I'm not a specialist in such imaging studies, the investigators seem to have performed well-designed experiments, analyzed rigorously. Some might complain that the paper is "purely descriptive," which it is, but I think it merits publication in the Journal of Cell Science because the results are so unexpected and they have substantial implications for areas such as thymic mimetic cells and development of thymus organoids.

I have just two minor suggestions:

- * The title could be changed to better convey the interest.
- * The authors could better discuss how they think this structure might relate to recently described thymic mimetic cells.

Reviewer 2

SUMMARY OF THE ADVANCE MADE IN THIS PAPER AND ITS POTENTIAL SIGNIFICANCE TO THE FIELD

While thymic cysts have been hypothesized to represent developmental remnants, the authors have provided compelling evidence that challenges this view by demonstrating the presence of an intrathymic respiratory element composed of epithelial cells resembling respiratory epithelium.

SUGGESTIONS TO AUTHORS

Although the authors do not clearly specify whether these epithelial cells are thymic epithelial cells or not, their findings raise the possibility that the recently identified mimetic TECs may include a subset distinct from thymic epithelial cells. As Krt5 and Krt8 used as epithelial markers in the manuscript are common between TECs and respiratory epithelium. Therefore, it is important to determine whether these respiratory epithelial cells within the thymus express markers specific to thymic epithelium. For example, antibodies specific for b5t, Ly51, or MHC-II molecules could serve as informative TEC markers. Alternatively, the authors should include a discussion on this point.

Furthermore, understanding the relationship between the intrathymic respiratory element and T cell development may be important for elucidating its biological significance. It would be informative to know whether this structure forms in the absence of mature thymocytes. For instance, is the respiratory element detectable in the fetal thymus, where the majority of thymocytes remain at CD4-CD8- stage? Alternatively, the authors should include a discussion on this point.

The genetic background, age, and sex of mice used in the study should be explicitly stated.

Reviewer 3

SUMMARY OF THE ADVANCE MADE IN THIS PAPER AND ITS POTENTIAL SIGNIFICANCE TO THE FIELD

The basic concept of this study is rather innovative and intriguing. Based on thorough histological analysis the study demonstrates that certain unusual thymic structures including isolated cysts or ducts are in fact, interconnected large epithelial networks (which the authors call thymic respiratory network) traversing from the capsule to the deep medulla. Furthermore, these epithelial networks are decorated by various thymic mimetic cells including ciliated, goblet, microfold, and ionocyte-like cells. In addition, the authors describe a morphologically unusual cell type characterized by large intracellular lumens lined with inward-facing cilia.

While these are interesting findings, in its current form, the manuscript would benefit from substantial revision to improve clarity, address unsupported assumptions, and strengthen the data interpretation. Below are my detailed comments and suggestions for improvement:

SUGGESTIONS TO AUTHORS

Major Comments

1. "Respiratory Network" and the proposed functions of the respiratory elements: The authors use the term "respiratory network" throughout the manuscript. Without direct evidence for respiratory-like function, this terminology is premature and potentially misleading. More cautious language would be advisable. Previous studies have shown that thymic mimetics can be broadly segregated into 3 major tissues: a) neuroendocrine (neuro-endocrine TEC mimetics); b) muscles (Myo-TEC mimetics) and c) epithelial/mucosal barriers (corneocytes, ionocytes, M-cells, tuft cells, cilia TEC, entero-TEC, etc....). The observed epithelial networks may thus rather resemble general organization of epithelial barriers rather than respiratory network - a highly misleading term. I urge the authors to reconsider their terminology, which I find inaccurate and misleading. Similarly, the manuscript suggests distinct functional roles for respiratory epithelial cells versus mimetic epithelial cells (e.g., Page 9, Line 3), but the proposed function of these structures remains purely speculative. If such functional distinctions are to be made, they must be supported with appropriate experimental data or clearly presented as hypotheses. In my opinion these represent mimetic cells that are not scattered randomly but rather have an organized pattern, similar to the peripheral barrier tissues. Correspondingly, I did not understand the following statement "Given its relative location to the medulla, it is highly likely that DCLK1+ tuft cells within intrathymic respiratory networks comprise a respiratory tuft cell population rather than a medullary mimetic subset." The authors should omit such baseless and misleading speculations, which unnecessarily weaken their study.

2. Characterization of mimetic cell types without appropriate markers
While the authors suggest cell identities based on morphology, these claims should be substantiated with immunostaining using known cell-type-specific markers, rather than relying solely on morphological resemblance to cells in other tissues. Some cell types are identified (e.g., DN cells in Figure 6a) without the use of specific molecular markers. These designations should either be supported by appropriate staining or revised to avoid definitive identification. Additionally, a summary figure or table listing the different mimetic cell types found in each region of the "thymic respiratory elements" would be highly useful.

3. Speculative Assumptions about Thymocyte Interaction
Several claims about the structural and functional role of these networks in thymocyte migration or egress (e.g., Page 11, Line 11, "tentacles as an entryway") seem over-interpreted and lack mechanistic support. It is unclear how different regions of the same structure (e.g., head vs. tentacles) would serve distinct roles unless they are composed of different cell types. 3D imaging data, if available, could help address these important structural-functional questions. (Note: I was unable to open the video files if they contain relevant data, please ensure they are accessible.)

4. Clarity and Rationale for Using Involved Thymi
The manuscript does not clearly explain the rationale for examining chemically involved thymi. A better justification is required, along with a clearer comparison between involved and normal thymi. Ideally, side-by-side images of both conditions should be included to facilitate the reader's appreciation of the differences and relevance to the study.

5. Lack of Experimental Details
There is no information provided regarding the age or sex of the mice used. These factors are critical, as the presence or morphology of these thymic structures may vary across developmental or aging stages. Please specify whether these features are consistent in young, adult, or old animals.

Minor Comments

- * The manuscript would benefit from a significant reduction in length, particularly in the figure descriptions, which are overly detailed and impede the flow of the text.
- * Some electron microscopy images are difficult to interpret. Highlighting or outlining select cell types in these images would make them much more informative and reader-friendly.

First revision

Author response to reviewers' commentsResponse to Reviewer Comments:Reviewer 1:

1. This paper is reporting unexpected, quite fascinating results. While I'm not a specialist in such imaging studies, the investigators seem to have performed well-designed experiments, analyzed rigorously. Some might complain that the paper is "purely descriptive," which it is, but I think it merits publication in the Journal of Cell Science because the results are so unexpected, and they have substantial implications for areas such as thymic mimetic cells and development of thymus organoids.

Response: We thank the reviewer for the positive comments.

2. I have just two minor suggestions:

*The title could be changed to better convey the interest.

*The authors could better discuss how they think this structure might relate to recently described thymic mimetic cells.

Response: Thank you for these suggestions. We have revised the title to: "Beyond Cysts: Organization of Structured Epithelial Networks in the Murine Thymus", which we believe is more engaging for readers. We opted to omit the term "respiratory" in order to address concerns raised by Reviewer 3 regarding the lack of definitive evidence establishing a respiratory identity for these cystic networks. Additionally, we expanded the conclusion section, to thoroughly discuss the relevance of our findings in light of recently described thymic mimetic cells. In particular, we now reference a new article from the Boehm group, published in *Nature* earlier this year (between our original and revised submissions), as also requested by both the editor and Reviewer 3.

Reviewer 2:

1. While thymic cysts have been hypothesized to represent developmental remnants, the authors have provided compelling evidence that challenges this view by demonstrating the presence of an intrathymic respiratory element composed of epithelial cells resembling respiratory epithelium. Although the authors do not clearly specify whether these epithelial cells are thymic epithelial cells or not, their findings raise the possibility that the recently identified mimetic TECs may include a subset distinct from thymic epithelial cells. Krt5 and Krt8 used as epithelial markers in the manuscript are common between TECs and respiratory epithelium. Therefore, it is important to determine whether these respiratory epithelial cells within the thymus express markers specific to thymic epithelium. For example, antibodies specific for b5t, Ly51, or MHC-II molecules could serve as informative TEC markers. Alternatively, the authors should include a discussion on this point.

Response: Thank you for the insightful recommendation. In response, we have added a discussion point in the fourth paragraph of the Conclusion section that explicitly addresses the limitation of not including a broader panel of TEC and mimetic markers. We also emphasize the importance of this analysis for future studies aiming to further dissect the identity and heterogeneity of the epithelial lining cells.

2. Furthermore, understanding the relationship between the intrathymic respiratory element and T cell development may be important for elucidating its biological significance. It would be informative to know whether this structure forms in the absence of mature thymocytes. For instance, is the respiratory element detectable in the fetal thymus, where the majority of thymocytes remain at CD4-CD8- stage? Alternatively, the authors should include a discussion on this point.

Response: We appreciate this important suggestion. Whether the formation of intrathymic respiratory elements depends on the presence or maturation of thymocytes remains an open question. Although our study did not include fetal thymi or models of T cell deficiency, we agree that investigating the presence and organization of these structures in such contexts, particularly in CD4-CD8- dominant environments, will be essential to understanding whether their development is thymocyte-independent or shaped by thymocyte-stromal interactions. We have now included this point in the conclusion section, as the fifth paragraph.

3. The genetic background, age, and sex of mice used in the study should be explicitly stated.

Response: This issue was also raised by Reviewer 3, and we apologize for the initial omission of this critical information. We have now addressed this by adding the relevant details in the Materials and Methods section under “Mouse Model of Cyclophosphamide-Induced Thymic Involution.”

Reviewer 3:

1. The basic concept of this study is rather innovative and intriguing. Based on thorough histological analysis the study demonstrates that certain unusual thymic structures including isolated cysts or ducts are in fact interconnected large epithelial networks (which the authors call thymic respiratory network) traversing from the capsule to the deep medulla. Furthermore, these epithelial networks are decorated by various thymic mimetic cells including ciliated, goblet, microfold, and ionocyte-like cells. In addition, the authors describe a morphologically unusual cell type characterized by large intracellular lumens lined with inward-facing cilia. While these are interesting findings, in its current form, the manuscript would benefit from substantial revision to improve clarity, address unsupported assumptions, and strengthen the data interpretation. Below are my detailed comments and suggestions for improvement:

Response: We thank the reviewer for acknowledging the importance of these data. We are now addressing and implementing point-by-point all their recommendations, to improve clarity and address unsupported assumptions. We hope that the manuscript is much improved at its revised state.

2. "Respiratory Network" and the proposed functions of the respiratory elements:

(a) The authors use the term "respiratory network" throughout the manuscript. Without direct evidence for respiratory-like function, this terminology is premature and potentially misleading. A more cautious language would be advisable. Previous studies have shown that thymic mimetics can be broadly segregated into 3 major tissues: a) neuroendocrine (neuro-endocrine TEC mimetics); b) muscles (Myo-TEC mimetics) and c) epithelial/mucosal barriers (corneocytes, ionocytes, M-cells, tuft cells, cilia TEC, entero-TEC, etc.). The observed epithelial networks may thus resemble general organization of epithelial barriers rather than respiratory network - a highly misleading term. I urge the authors to reconsider their terminology, which I find inaccurate and misleading.

Response: We thank the reviewer for suggesting a revision in terminology and nomenclature. This is indeed a complex issue, as earlier studies have consistently referred to these epithelial structures as “respiratory cysts,” “tubules,” and related terms, based on their clear morphological resemblance to respiratory cell types. However, we recognize that the reviewer’s recommendation aligns with more recent nomenclature, informed by advanced approaches such as single-cell RNA sequencing, which offer a refined understanding of thymic epithelial cell heterogeneity.

A key challenge in the field is the lack of a conceptual framework that bridges historical morphological observations with emerging molecular classifications. One of the goals of our interpretative study was to address this gap. In light of the reviewer’s helpful suggestion, we have revised the manuscript accordingly. Throughout the text, from the title to the conclusions, we have removed references to these structures as “respiratory” and now refer to them more broadly as “epithelial,” consistent with the reviewer’s input. Approximately ~120 such instances were corrected throughout the manuscript, including the supplementary files.

Nevertheless, in a few sections, we discuss how our current findings relate to earlier descriptions of “respiratory elements.” In these cases, we clearly indicate that we are offering a justified and well-supported hypothesis: that these epithelial networks likely consist of cell types resembling respiratory epithelia, as a means to integrate past and present perspectives.

(b) Similarly, the manuscript suggests distinct functional roles for respiratory epithelial cells versus mimetic epithelial cells (e.g., Page 9, Line 3), but the proposed function of these structures remains purely speculative. If such functional distinctions are to be made, they must be supported with appropriate experimental data or clearly presented as hypotheses. In my opinion these represent mimetic cells that are not scattered randomly but rather have an organized pattern, similar to the peripheral barrier tissues.

Response: We completely agree with the reviewer’s interpretation that the respiratory-like cells

within these networks likely represent mimetic epithelial populations arranged in an organized pattern, rather than being randomly distributed. While this is an intriguing and important hypothesis, a thorough experimental investigation would require a dedicated and complex study, which we believe should be pursued in future work. Given that such analyses are beyond the scope of the current manuscript, we have revised the text to clearly frame these interpretations as speculative and explicitly presented as hypotheses. This ensures that we avoid overinterpretation and maintain a cautious and appropriate tone throughout the manuscript.

Two key instances, one of which was directly suggested by the reviewer, where we implemented such changes are: (i) the final paragraph on page 10, where we explicitly state that the proposed origin of SP cells within the cavities of epithelial networks is a hypothesis; and (ii) the first paragraph on page 15, where we now clearly define the idea that respiratory-like cells may represent mimetic populations as a testable hypothesis.

(c) Correspondingly, I did not understand the following statement "Given its relative location to the medulla, it is highly likely that DCLK1⁺ tuft cells within intrathymic respiratory networks comprise a respiratory tuft cell population rather than a medullary mimetic subset." The authors should omit such baseless and misleading speculations, which unnecessarily weaken their study.

Response: Thank you for the recommendation. This confusing sentence is entirely omitted.

3. Characterization of mimetic cell types without appropriate markers:

(a) While the authors suggest cell identities based on morphology, these claims should be substantiated with immunostaining using known cell-type-specific markers, rather than relying solely on morphological resemblance to cells in other tissues. Some cell types are identified (e.g., DN cells in Figure 6a) without the use of specific molecular markers. These designations should either be supported by appropriate staining or revised to avoid definitive identification.

Response: Thank you for this recommendation. While there is well-established textbook knowledge on distinguishing DN from DP thymocytes using electron microscopy, i.e., knowledge that is foundational for pathologists and microscopists and typically does not require citation, we appreciate that identifying cell populations without definitive lineage markers may be confusing for readers, particularly as our understanding of thymocyte subsets continues to evolve. We acknowledge this limitation and have revised the text to clarify the interpretative nature of these identifications. As such, we made the following changes:

- i.* We have revised Fig 6 by removing all designations of thymocytes as DN cells. In the accompanying text and figure legend, we now describe these cells more cautiously as thymocytes with abundant cytoplasm ("large thymocytes"), which may suggest an earlier stage of thymopoiesis rather than later ones, i.e., DP or SP. However, we present this interpretation as speculative and therefore have taken care to avoid overinterpretation bias.
- ii.* In addition to the specific example noted by the reviewer, we have now addressed a broader limitation of the study: this is fundamentally an imaging- and morphology-based investigation, relying on the structural and phenotypic resemblance of cells. While the identification of cell types involved veterinary pathologists, microscopists, and thymus biologists, we acknowledge that the absence of lineage marker staining in most cases limits definitive classification. As such, the data should be interpreted with caution. Exceptions to this limitation are explicitly noted in the manuscript, for example, when DCLK1 and AIRE were used to identify specific mTEC subsets. This discussion is shown in the study's limitations in the conclusion section (page 20; second to last paragraph).

We believe these refinements adequately address the reviewer's concerns. However, if there are any additional specific instances where the reviewer recommends revising the interpretation of particular cell identifications, we would be happy to consider and incorporate those suggestions.

(b) Additionally, a summary figure or table listing the different mimetic cell types found in each region of the "thymic respiratory elements" would be highly useful.

Response: This is a great idea. We have created Table 2 to summarize these findings.

4. Speculative Assumptions about Thymocyte Interaction:

(a) Several claims about the structural and functional role of these networks in thymocyte migration or egress (e.g., Page 11, Line 11, "tentacles as an entryway") seem overinterpreted and lack mechanistic support. It is unclear how different regions of the same

structure (e.g., head vs. tentacles) would serve distinct roles unless they are composed of different cell types.

Response: We thank the reviewer for raising this important point. We agree that, in the absence of direct mechanistic evidence, any proposed functional roles for specific regions of the epithelial networks, such as the suggestion that Tentacles may serve as an egress route, must be treated with caution. We would like to clarify, however, that in many biological systems, including the respiratory mucosa, regional functional differences are often conferred not by abrupt changes in cell types, but by shifts in the proportions of shared cell populations. For example, ciliated cells are present throughout the proximal and distal airways, but their abundance progressively decreases toward the alveoli, where they are ultimately absent. These proportional changes in epithelial composition contribute to key regional functions—such as the mucociliary escalator—without requiring entirely distinct cell types. Similarly, secretory cell subtypes also vary in proportion along the airway tract to support local tissue needs.

By analogy, the structural heterogeneity observed across the Head, Neck, Funnel, and Tentacle portions of the intrathymic epithelial networks may reflect changes in the relative abundance of shared epithelial populations, rather than discrete compartmentalization. Nonetheless, as the thymus is a unique immune organ and these epithelial structures are putative mimetic forms rather than true respiratory mucosa, their precise functions remain unclear.

In agreement with the reviewer, we have therefore removed speculative statements suggesting defined interactions between epithelial cells and thymocytes within these networks. While we retain some explanatory discussion regarding the presence of thymocytes in the lumen, framed as a hypothesis, such interpretations are now explicitly identified as speculative to avoid overstatement. To directly address the issue of tentacles treated as entryways in the absence of kinetic or mechanistic data we have now:

- i. Removed this statement from the abstract.
- ii. Added an explicit sentence in Page 12, discussing that the alternative thymocyte egress pathway is only a speculative model, but explains phenotypic findings in our current study, and past literature.

We believe that by raising the concern to readers, and keeping reminding them that the study is descriptive and phenotypic, we avoid overstatement biases, but at the same time refine the new hypotheses and possibilities that might explain how mimetic cells might be educating thymocytes

(b) 3D imaging data, if available, could help address these important structural-functional questions. (Note: I was unable to open the video files, if they contain relevant data, please ensure they are accessible.)

Response: We agree with the reviewer that three-dimensional reconstructions would be invaluable for elucidating the full architecture and complexity of these epithelial networks. However, the networks are extremely convoluted and expansive, such that meaningful insights would likely require reconstruction at the scale of the entire organ. The supplemental data that the reviewer was unable to access pertain to a solitary ciliated TEC, for which we were able to perform partial or complete single-cell reconstructions. Unfortunately, this does not extend to the larger epithelial networks, which are likely to comprise hundreds to thousands of interconnected epithelial cells. We hope that our study can serve as a foundational framework for future investigations of the type suggested by the reviewer, potentially incorporating whole-organ 3D reconstruction technologies to fully resolve the topology and organization of these remarkable structures.

We provided the video files in widely supported and distributable formats to ensure broad accessibility. We anticipate that the editors will provide feedback and confirmation that the files are viewable on the reviewer's or reader's system, and we are happy to provide alternative formats if needed to ensure smooth playback.

5. Clarity and Rationale for Using Involved Thymi:

The manuscript does not clearly explain the rationale for examining chemically-involved thymi. A better justification is required, along with a clearer comparison between involved and normal thymi. Ideally, side-by-side images of both conditions should be included to facilitate the reader's appreciation of the differences and relevance to the study.

Response: Thank you for this recommendation. We have now added a small paragraph (3rd) and modified the last paragraph (4th) in the introduction, to expand on the rationale for examining chemically-involved thymi.

6. Lack of Experimental Details:

There is no information provided regarding the age or sex of the mice used. These factors are critical, as the presence or morphology of these thymic structures may vary across developmental or aging stages. Please specify whether these features are consistent in young, adult, or old animals.

Response: We apologize for omitting this critical information. As also noted by Reviewer 2 (Comment 3), we have now included the relevant details in the Materials and Methods section under “Mouse Model of Cyclo- phosphamide-Induced Thymic Involution.”

7. Minor Comments:

***The manuscript would benefit from a significant reduction in length, particularly in the figure descriptions, which are overly detailed and impede the flow of the text.**

***Some electron microscopy images are difficult to interpret. Highlighting or outlining select cell types in these images would make them much more informative and reader-friendly.**

Response: We appreciate these minor comments aimed at improving the manuscript. In response to the reviewers’ collective suggestions, we have reduced the overall length by approximately 300 words, primarily by trimming overly detailed descriptions. Additionally, we enhanced the clarity of the electron microscopy images by adding more tags to guide readers through the relevant structures. However, we were careful not to tag or label any cell unless the team of specialists among the authors was confident in its identification, in order to maintain accuracy and avoid misinterpretation.

As a last note, we would like to confirm that the revised manuscript, has addressed certain editorial issues, such as presenting the references in an acceptable format, reducing the abstract size, and adding specific elements in the revised document.

Second decision letter

MS ID#: jcs.264079R1

MS Title: Beyond Cysts: Organization of Structured Epithelial Networks in the Murine Thymus

Authors: Stepan Vodopyanov; Leslie Gunther-Cummins; Sophia DesMarais; Xheni Nishku; Joseph Churaman; Hillary Guzik; Rotem Alon; Vera DesMarais; Frank Macaluso; George Karagiannis
Article Type: Research Article

Dear George and Stepan,

I am pleased to inform you that your manuscript has been accepted for publication in our Journal of Cell Science Special Issue on Cilia and Flagella, pending standard publication integrity checks. It was accepted on 11 Aug 2025. Where referee reports on this version are available, they are appended below.